# Homeostatic scaling is driven by a translation-dependent degradation axis that recruits miRISC remodeling

**Balakumar Srinivasan**[1ʘ], **Sarbani Samaddar**[1ʘ], **Sivaram V. S. Mylavarapu**[2], **James P. Clement**[3], **Sourav Banerjee**[1]*

**1** National Brain Research Centre, Manesar, India, **2** Regional Centre for Biotechnology, Faridabad, India, **3** Neuroscience Unit, Jawaharlal Nehru Centre for Advanced Scientific Research, Bengaluru, India

ʘ These authors contributed equally to this work.
* souravnbrc@gmail.com; sourav@nbrc.ac.in

**Data Availability Statement:** https://figshare.com/articles/dataset/Homeostatic_scaling_is_driven_by_a_translation-dependent_degradation_axis_that_recruits_miRISC_remodeling/16768816.

## Abstract

Homeostatic scaling in neurons has been attributed to the individual contribution of either translation or degradation; however, there remains limited insight toward understanding how the interplay between the two processes effectuates synaptic homeostasis. Here, we report that a codependence between protein synthesis and degradation mechanisms drives synaptic homeostasis, whereas abrogation of either prevents it. Coordination between the two processes is achieved through the formation of a tripartite complex between translation regulators, the 26S proteasome, and the miRNA-induced silencing complex (miRISC) components such as Argonaute, MOV10, and Trim32 on actively translating transcripts or polysomes. The components of this ternary complex directly interact with each other in an RNA-dependent manner. Disruption of polysomes abolishes this ternary interaction, suggesting that translating RNAs facilitate the combinatorial action of the proteasome and the translational apparatus. We identify that synaptic downscaling involves miRISC remodeling, which entails the mTORC1-dependent translation of Trim32, an E3 ligase, and the subsequent degradation of its target, MOV10 *via* the phosphorylation of p70 S6 kinase. We find that the E3 ligase Trim32 specifically polyubiquitinates MOV10 for its degradation during synaptic downscaling. MOV10 degradation alone is sufficient to invoke downscaling by enhancing Arc translation through its 3′ UTR and causing the subsequent removal of postsynaptic AMPA receptors. Synaptic scaling was occluded when we depleted Trim32 and overexpressed MOV10 in neurons, suggesting that the Trim32-MOV10 axis is necessary for synaptic downscaling. We propose a mechanism that exploits a translation-driven protein degradation paradigm to invoke miRISC remodeling and induce homeostatic scaling during chronic network activity.

## Introduction

Neurons employ a unique strategy, known as synaptic scaling, to counter the runaway excitation and subsequent loss of input specificity that arise due to Hebbian changes; they rely on a

**Funding:** This work was supported by Ramalingaswami Fellowship (BT/RLF/Re-entry/32/2011) from the Department of Biotechnology, Government of India (www.dbtindia.gov.in) and core funding by the National Brain Research Centre (www.nbrc.ac.in) to S.B. The funders had no role in study design, data collection and analysis, decision to publish, or preparation of the manuscript.

**Competing interests:** The authors have declared that no competing interests exist.

**Abbreviations:** Ago, Argonaute; AMPAR, AMPA receptor; DIV, days in vitro; HA, haemagglutinin; mEPSC, miniature excitatory postsynaptic current; miRISC, miRNA-induced silencing complex; mTORC1, mammalian Target Of Rapamycin Complex-1; PSD, postsynaptic density; p70 S6K, p70 S6 kinase; sAMPAR, surface AMPAR; SD, Sprague Dawley; UPS, ubiquitin proteasome system; 4E-BP2, 4E-binding protein 2.

compensatory remodeling of synapses throughout the network while maintaining differences in their synaptic weightage [1–6]. Synaptic scaling is achieved by a complex interplay of sensors and effectors within neurons that serve to oppose global fluctuations in a network and establish synaptic homeostasis by modifying postsynaptic glutamatergic currents in a cell-autonomous manner [7–9]. In the context of homeostatic scaling, "sensors" are classified as molecules that sense deviations in the overall network activity, and "effectors" scale the neuronal output commensurately.

Till date, not much is known about the repertoire of molecular "sensor" cascades that serve to link events where neurons sense deviations in the network firing rate and subsequently initiate the scaling process. Few molecular sensors have been identified; the eukaryotic elongation factor eEF2 and its dedicated kinase, eEF2 kinase or CamKIII, are the two reported thus far [10]. One cascade discovered in this context is the mammalian Target Of Rapamycin Complex-1 (mTORC1) signaling pathway that regulates presynaptic compensation in neurons by promoting BDNF synthesis in the postsynaptic compartment [11,12]. In contrast, AMPA receptors (AMPARs) have been identified, by overwhelming consensus, to be the predominant "end-point-effectors" in all paradigms of synaptic scaling [13–16]. Unlike NMDARs, AMPARs undergo *de novo* translation during network destabilizations [17], and chronic changes in the postsynaptic response during scaling have been attributed to the abundance of surface AMPARs (sAMPARs) (GluA1 and GluA2 subunits) [18]. Among the key modifiers of AMPAR expression, miRNAs are known to play pivotal roles in synaptic scaling [19–22]. Relief from translational repression by miRNAs necessitates that mRNAs exit the functional microRNA-induced silencing complex (miRISC). This requires miRISC to undergo dynamic changes in its composition [23,24], a cellular phenomenon previously termed as miRISC remodeling [25]. However, what remains surprising is our lack of knowledge about how compositional changes within the miRISC are achieved during scaling.

The requirement for discrete sets of sensors and effectors is fulfilled within neurons through varied mechanisms including translation and ubiquitin proteasome system (UPS) degradation. An enhanced degradation of postsynaptic density (PSD) proteins including GluA1 and GluA2 has been observed in contexts of altered network excitability [26], whereas complete inhibition of UPS activity was shown to occlude synaptic compensation [27]. The integral role of *de novo* translation in synaptic homeostasis was recently highlighted when proteomic analysis of neurons undergoing upscaling and downscaling revealed a remarkable diversity of newly synthesized proteins. Of particular interest was the significant enrichment in the expression of the proteasome core complex during downscaling [28,29]. The demand for the translation of proteasome complexes implies that proteasomes work alongside translation mechanisms during downscaling. Reports documenting the colocalization of ribosomes and the proteasome in neuronal dendrites [30,31] further emphasize the possibility that these two opposing machineries physically interact within the postsynaptic compartment. The remodeling of the proteome through the dynamic regulation of protein biogenesis and degradation has been termed as cellular "proteostasis" [32]. However, several questions remain unexplored in the context of cellular proteostasis during homeostatic scaling, such as (a) What factor establishes the link between translation and protein degradation machineries to shape the proteome during scaling? (b) Which process among translation and degradation takes precedence? (c) What are the signaling mechanisms that connect events of "sensing" the bicuculline-mediated hyperactivity and the final down-regulation of sAMPARs?

Here, we demonstrate a defined mechanism of synaptic scaling accomplished through an RNA-dependent coordination between translation and proteasome-mediated degradation. We observe that isolated inhibition of either translation or proteasomal activity offsets synaptic homeostasis. Restoration of homeostasis necessitates the combination of both processes. We

provide empirical evidence demonstrating that the interaction between translation and protein degradation machineries is direct and RNA dependent. This coordination is achieved when the two apparatuses are tethered to actively translating transcripts linked to miRISC. Synaptic hyperactivity causes an increased abundance of Trim32 and depletion of MOV10 in polysomes; both Trim32 and MOV10 are members of the miRISC. We find that in contexts of chronic hyperactivity, mTORC1-dependent translation of the E3 ligase Trim32 promotes the polyubiquitination and subsequent degradation of MOV10 by proteasome. This is triggered by the mTORC1-mediated phosphorylation of its downstream effector, p70 S6 kinase (p70 S6K). We observe that MOV10 degradation leads to enhanced translation of Arc and results in the reduced distribution of sAMPARs. Loss of MOV10 alone is sufficient to decrease the synaptic strength by reducing sAMPARs and mimic events similar to hyperactivity-driven downscaling. Notably, the observed increase in Arc expression in the context of synaptic downscaling happens *via* translation and not by transcriptional mechanisms.

## Results

### Codependence of protein synthesis and degradation drives synaptic homeostasis

To test the existence of coordination between translation and degradation in the regulation of synaptic homeostasis, we measured miniature excitatory postsynaptic currents (mEPSCs) from cultured hippocampal neurons (days in vitro (DIV) 18 to 24) after pharmacological inhibition of protein synthesis (anisomycin, 40 μM) and proteasomal activity (lactacystin, 10 μM) for 24 hours. Application of either lactacystin or anisomycin increased (2.43 ± 0.43 pA, $p < 0.02$) and decreased (5.86 ± 0.13 pA, $p < 0.01$) mEPSC amplitude respectively. Coapplication of both inhibitors restored mEPSC amplitude to that of vehicle treated neurons (Fig 1A and 1B). The frequency of mEPSCs remained unaltered upon inhibition of translation and proteasome blockade either alone or in combination (Fig 1C), suggesting that this could be a postsynaptic phenomenon. Our data imply that interfering with either protein synthesis or degradation disturbs the balance of synaptic activity, while blocking both synthesis and degradation altogether restores it. Next, we stimulated synaptic downscaling using bicuculline (10 μM, 24 hours) and observed that, like previous reports, here, too, chronic application of bicuculline leads to a significant decrease in mEPSC amplitude (5.70 ± 0.08 pA, $p < 0.01$) without any detectable change in frequency (Fig 1D, 1E and 1F). The extent of decrease in mEPSC amplitude within bicuculline-treated neurons recapitulated the decrease observed in neurons where translation was blocked (bicuculline treated neuron 5.70 ± 0.08 pA decrease versus anisomycin-treated neuron 5.86 ± 0.13 pA decrease) (Fig 1B and 1E). We measured the mEPSC amplitude and frequency from hippocampal neurons when bicuculline was coapplied with anisomycin and lactacystin. The dual application of bicuculline and anisomycin did not result in any significant change in mEPSC amplitude when compared to neurons treated with bicuculline alone (Fig 1D and 1E). This confirms that, rather than inducing an additive effect, chronic inhibition of protein synthesis in itself is sufficient to induce downscaling and could potentially override the effect observed due to bicuculline. Disruption of proteasome function by lactacystin during bicuculline treatment led to a significant increase in mEPSC amplitude (9.46 ± 0.07 pA increase as compared to bicuculline treated neurons, $p < 0.001$) without altering frequency (Fig 1E and 1F). The increase was effectively more than the basal activity of vehicle-treated neurons (3.76 ± 0.08 pA, $p < 0.01$) and mimicked the increase in mEPSC amplitude brought by lactacystin alone (Fig 1B and 1E). Although the influence of lactacystin on mEPSC amplitude is opposite to that of anisomycin, their individual effects override that of bicuculline in each condition. Coapplication of both inhibitors during bicuculline-induced

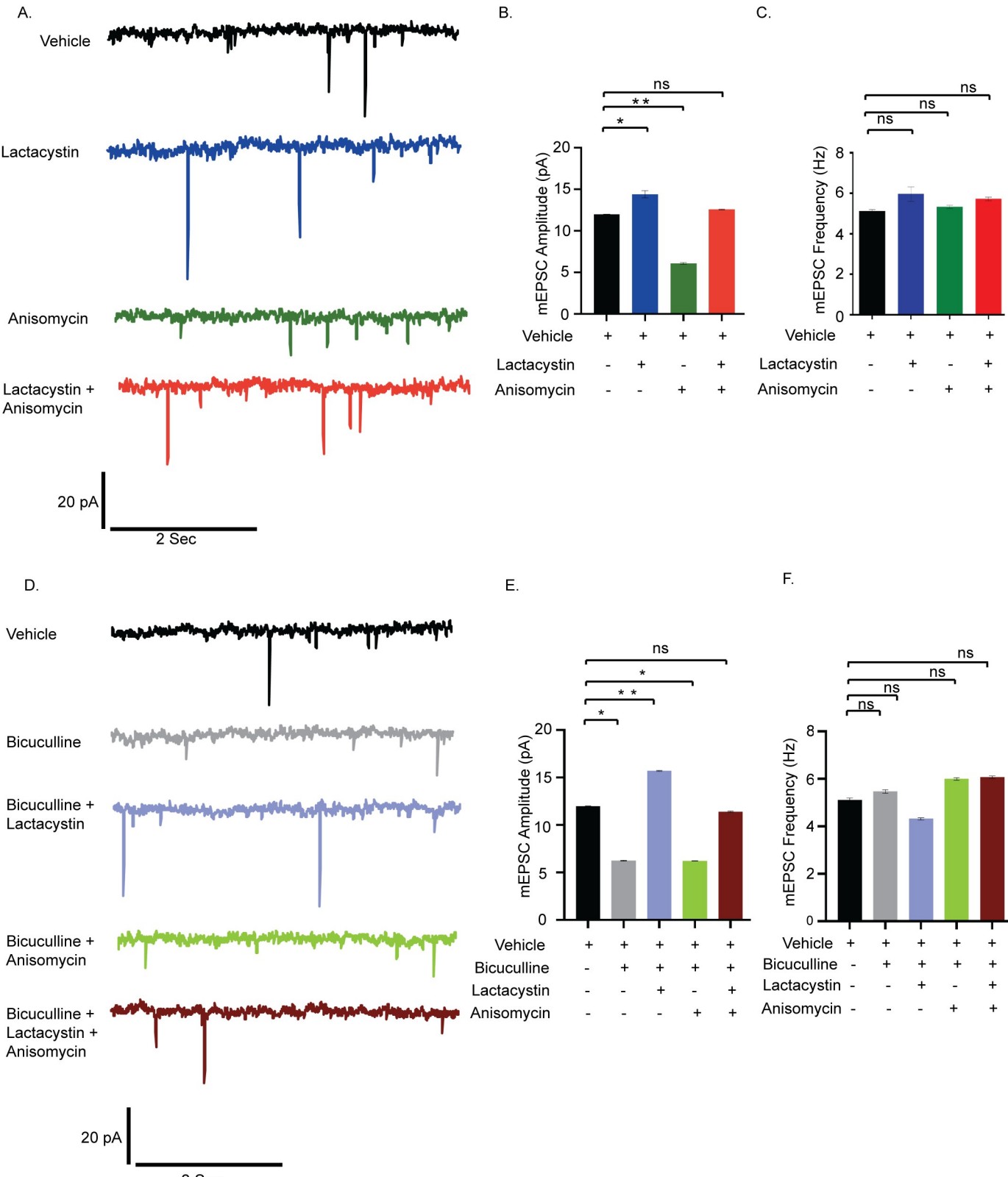

**Fig 1. Synaptic scaling is coregulated by protein synthesis and degradation. (A)** mEPSC traces from hippocampal neurons treated with vehicle, lactacystin, anisomycin, and both. **(B)** Mean mEPSC amplitude. **(C)** Mean mEPSC frequency. $n = 13$–$15$. $^*p < 0.024$, $^{**}p < 0.01$. ns, not significant. Data shown as

mean ± SEM. One-way ANOVA and Fisher's LSD. Scale as indicated. **(D)** mEPSC traces from neurons treated with vehicle, bicuculline alone, or in combination with lactacystin and anisomycin. **(E)** Mean mEPSC amplitude. **(F)** Mean mEPSC frequency. $n = 12$–$16$. $^*p < 0.01$, $^{**}p < 0.001$. ns, not significant. Data shown as mean ± SEM. One-way ANOVA and Fisher's LSD. Scale as indicated. The data underlying this figure are available at https://figshare.com/articles/dataset/ Homeostatic_scaling_is_driven_by_a_translation-dependent_degradation_axis_that_recruits_miRISC_remodeling/16768816. mEPSC, miniature excitatory postsynaptic current; ns, not significant.

hyperactivation produced mEPSC amplitudes comparable to vehicle-treated neurons (Fig 1E). Our data indicate that the coinhibition of translation and degradation restricts any molecular changes away from the basal level, thus maintaining the synaptic strength at the established physiological set point.

## Synchronized translation and degradation regulates AMPAR distribution during scaling

Since adjustment of synaptic strengths is directly correlated to the distribution of sAMPARs, we measured the surface expression of GluA1 and GluA2 (sGluA1/A2) to identify how concerted mechanisms of synthesis and degradation influence the distribution of sAMPARs during scaling. Neurons (DIV 21 to 24) were live labeled using N-terminus specific antibodies against GluA1 and GluA2 following bicuculline treatment, either alone or in presence of both anisomycin and lactacystin, for 24 hours and synapses marked by PSD95. The surface expression of sGluA1/A2 in excitatory neurons was decreased following network hyperactivity ($50.6 \pm 6.68\%$, $p < 0.01$ for sGluA1 and $26.1 \pm 6.62\%$, $p < 0.01$ for sGluA2) (Figs 2A–2D and S1A and S1B). Consistent with our electrophysiological data, inhibition of both the translation and the proteasome in bicuculline-treated neurons increased sGluA1/A2 levels ($133.95 \pm 8.77\%$, $p < 0.01$ for sGluA1, $53.17 \pm 6.44\%$, $p < 0.001$ for sGluA2) when compared to neurons treated with bicuculline alone (Figs 2C, 2D, S1A, and S1B). Thus, our data indicate that a dual inhibition of protein synthesis and degradation restores the synaptic sGluA1/A2 following network hyperactivity.

To reaffirm whether AMPARs are indeed the end-point effectors of synaptic downscaling, we used GluA2$_{3Y}$, a synthetic peptide derived from the GluA2 carboxy tail of AMPA receptors to block the endocytosis of the AMPARs [13], effectively ensuring that the number of sAMPARs remain unchanged throughout 24 hours. Consistent with previous studies, no significant changes in mEPSC amplitude were detected upon inhibition of GluA2 endocytosis during chronic application of bicuculline (GluA2$_{3Y}$-treated neuron $11.01 \pm 0.36$ pA versus GluA2$_{3Y}$ + bicuculline-treated neuron $12.17 \pm 0.28$ pA, $p < 0.49$) (Fig 2E and 2F).

Application of GluA2$_{3Y}$ did not alter mEPSC amplitude as compared to vehicle-treated neurons (GluA2$_{3Y}$-treated neuron $11.01 \pm 0.36$ versus vehicle-treated neurons $11.94 \pm 0.07$ pA) (S1C and S1D Fig), nor any change observed between neurons treated with GluA2$_{3Y}$ and those treated with both lactacystin and anisomycin in presence or absence of bicuculline (Fig 2F and 2G). mEPSC frequency remained unaltered throughout, while mEPSC amplitude in each condition was similar to that of control neurons (Figs 2G and S1E). Collectively, these observations indicate that changes in the abundance of sAMPARs during scaling is facilitated by proteomic remodeling that exploits both translation and degradation processes.

## RNA-dependent cosedimentation of the proteasome and translation regulators

The colocalization of polyribosomes and proteasomes to sites of synaptic activity [30,31] lead us to examine whether the components of the 26S proteasomal machinery could remain physically associated with actively translating transcripts in order to make the necessary proteomic

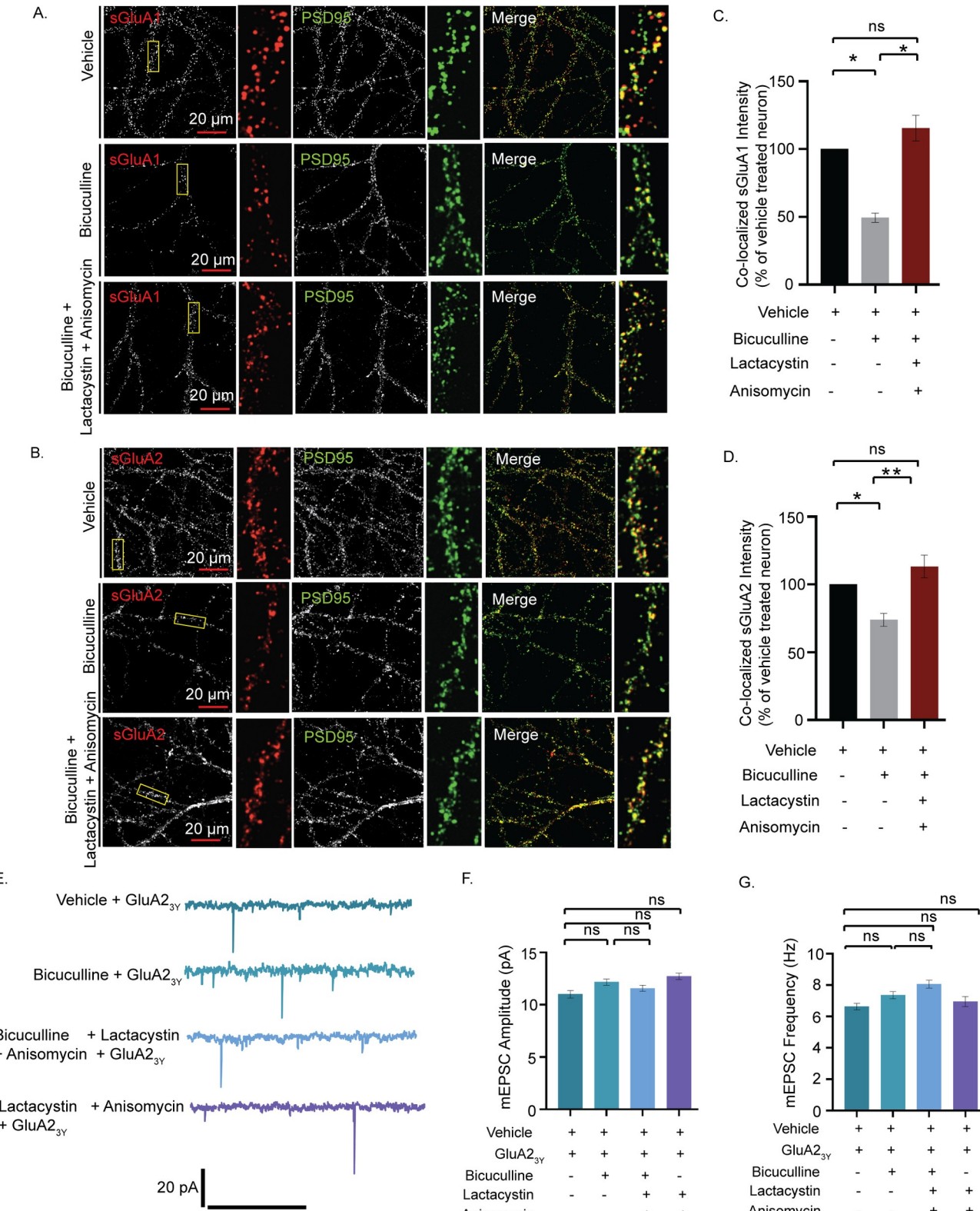

**Fig 2. Coinhibition of protein synthesis and degradation restores hyperactivity-driven reduction of synaptic AMPAR expression. (A)** High magnification images of sGluA1 (red), PSD95 (green), and sGluA1/PSD95 (merged) and **(B)** high magnification images of sGluA2 (red), PSD95 (green), and sGluA2/PSD95 (merged) from neurons treated with vehicle, bicuculline alone, or in combination with lactacystin and anisomycin. **(C)** Normalized intensity of sGluA1 colocalized with PSD95 particles.

**(D)** Normalized intensity of sGluA2 colocalized with PSD95 particles. $n = 56–57$, sGluA1 and $n = 31–63$, sGluA2. $^*p < 0.01$, $^{**}p < 0.001$. One-way ANOVA and Fisher's LSD. Dendrite marked in yellow box was digitally amplified. See also S1 Fig. **(E)** mEPSC traces from hippocampal neurons treated with $GluA_{23y}$ either alone or in presence of bicuculline, lactacystin + anisomycin, and bicuculline + lactacystin + anisomycin. **(F)** Mean mEPSC amplitude. **(G)** Mean mEPSC frequency. $n = 10–13$. ns, not significant. Data are shown as mean ± SEM. One-way ANOVA and Fisher's LSD. Scale as indicated. See also S1 Fig. The data underlying this figure are available at https://figshare.com/articles/dataset/Homeostatic_scaling_is_driven_by_a_translation-dependent_degradation_axis_that_recruits_miRISC_remodeling/16768816. AMPAR, AMPA receptor; mEPSC, miniature excitatory postsynaptic current; ns, not significant.

changes. These components include proteins forming the 19S regulatory subunits and the 20S proteasome core. We analyzed polysomes from the hippocampus of 8- to 10-week-old rats and assessed whether the sedimentation pattern of proteasomes match those of actively translating, polyribosome-associated mRNA fractions. We observed that several components of the proteasomal machinery such as α7 subunit of the 20S proteasome; and Rpt1, Rpt3, and Rpt6 subunits of the 19S proteasome cosedimented with translation initiation factors such as eIF4E and p70 S6K within actively translating polysomes (Figs 3A, 3B, and S2A). We also detected the polysomal distribution of MOV10, a helicase and an RNA-binding protein known to be polyubiquitinated upon synaptic activation, and Trim32, an E3 ligase, both components of the miRISC [23,33] (Fig 3A and 3B).

RNase or EDTA treatment of cytoplasmic lysates prior to density gradient fractionation led to a complete collapse of the polysome profile, simultaneously shifting the sedimentation of Rpt6, Rpt1, α7, eIF4E, MOV10, Trim32 to the lighter fractions (Figs 3C–3F, S2B, and S2C). The disruption of association between the translational and proteasomal components on RNase and EDTA treatment suggests that translating transcripts are necessary to recruit the translation and proteasome machineries. These observations ruled out the possibility that the observed cosedimentation was a result of similar densities between the protein complexes and polysomes. Trim32 and MOV10 in specific high-density sucrose fractions (fraction # 8/11/15) were not detected due to loss of proteins during the TCA precipitation step (Fig 3B). Furthermore, we saw that the polysome-associated 26S proteasome is catalytically active as detected by its ability to cleave a fluorogenic proteasome substrate that is blocked by the proteasome inhibitor epoxomicin (Figs 3G, 3H, and S2D).

## Proteasome and the regulators of translation directly interact with each other within excitatory neurons

Whole-cell patch clamp recordings from hippocampal excitatory neurons demonstrated that the coregulation of translation and proteasome-mediated protein degradation is necessary for synaptic homeostasis. Consistent with this observation, cosedimentation of proteasome subunits along with polysomes linked to protein synthesis regulators and members of the miRISC led us to enquire whether components of the ternary complex directly interact with each other in excitatory neurons of the hippocampus. To evaluate this, we immunoprecipitated the 19S proteasomal complex using Rpt6 antibody from hippocampal neurons. We observed the coprecipitation of eEF2, a translation elongation factor that functions as a "sensor" of change in network activity (Fig 4A). We also found that a known regulator of mTORC1-dependent protein synthesis; p70 S6K as well as its phosphorylated form [34] coprecipitated with the 19S proteasome (Fig 4A). We further analyzed the proteins interacting with polysomes within excitatory neurons by expressing haemagglutinin (HA)-tagged ribosomal protein Rpl22 (HA-Rpl22) that gets incorporated into polysomes [35,36] (Figs 4B–4D and S2E). We reasoned that the analysis of HA-Rpl22-affinity purified complexes would confirm whether the polysome-associated translation and degradation machineries directly interact with each other.

Our western blot analysis of HA-Rpl22 affinity-purified protein complex showed that Rpt6 directly interacts with Trim32 and MOV10 (Fig 4E). Immunoprecipitation of MOV10 from

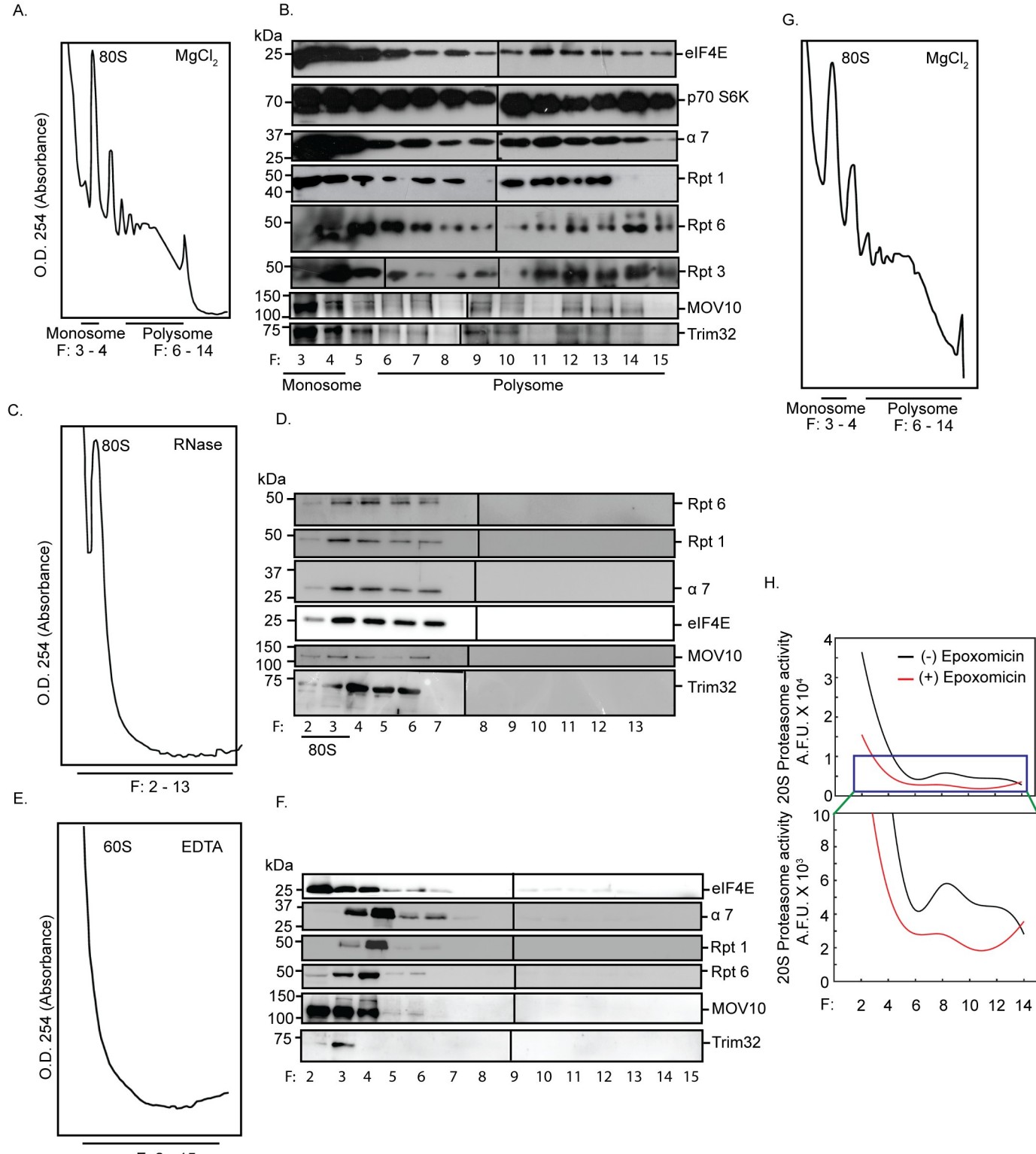

**Fig 3. RNA-dependent association between active proteasomes and translating polyribosomes.** Absorbance profile at 254 nm (A$_{254}$) and western blot analysis of fractionated cytoplasmic extracts from hippocampal tissue incubated in the absence or presence of MgCl$_2$ or RNase or EDTA. Monosome (80S), 60S ribosome, and polysome fractions are as indicated. Western blots performed from tri-chloroacetic acid–precipitated fractions to determine the distribution of the translation regulators eIF4E and p70 S6K; α7 subunit of the 20S core of the proteasome, Rpt1, Rpt3, Rpt6 of the 19S cap, and miRISC proteins MOV10 and Trim32 in the

presence or absence of MgCl$_2$ or RNase or EDTA. **(A)** A$_{254}$ profile in the presence of MgCl$_2$. **(B)** Western blots of the fractions obtained in (A). **(C)** A$_{254}$ profile obtained in the presence of RNase. **(D)** Western blots of the fractions obtained in (C). **(E)** A$_{254}$ profile in the presence of EDTA. **(F)** Western blots of the fractions obtained in (E). Rpt3 blots with different exposures are distinguished by a vertical black line to denote they represent separate panels within the figure. Two blots with different exposures are shown in the main figure and raw data to visualize the specific band of Rpt3. **(G)** A$_{254}$ profile of fractionated cytoplasmic extract used for determining activity of proteasomes. **(H)** Quantitation of catalytic activity of proteasomes present in alternate fractions from two polysome preparations. See also S2 Fig. The data underlying this figure are available at https://figshare.com/articles/dataset/Homeostatic_scaling_is_driven_by_a_translation-dependent_degradation_axis_that_recruits_miRISC_remodeling/16768816. miRISC, miRNA-induced silencing complex; p70 S6K, p70 S6 kinase.

hippocampal neurons resulted in the coprecipitation of both Argonaute (Ago) and Trim32, confirming that the latter is an integral component of the Ago-containing miRISC (Fig 4F). We also detected the chaperone protein HspA2 in the HA-affinity purified fraction along with Rpt6 (Fig 4G), suggesting that HspA2 could tether proteasomes to actively translating transcripts. The direct interaction between components of the translation and proteasome machinery could occur without the participation of polysome-associated, translating mRNA. To evaluate whether the observed association was RNA dependent or RNA independent, HA-Rpl22 affinity-purified protein complexes from polysome fractions of hippocampal tissue lysates treated with or without RNase were analyzed (Fig 4H, 4I, S2F and S2G Fig). Our western blot analysis revealed that the 20S proteasome core, Rpt6, and Rpt1 coprecipitated with eIF4E and p70 S6K (Fig 4J). RNase treatment of the cytoplasmic lysate prior to density gradient fractionations prevented this interaction on the actively translating, heavier fractions of the polysome (Fig 4J). This demonstrates that polysome-associated, translating RNA act as scaffolds to facilitate the direct interaction between protein synthesis and degradation modules.

## Chronic hyperactivity in neurons regulates the distribution of factors associated with translation and the proteasomal machinery

Once we identified that members of the translation apparatus, the 26S proteasome, and the miRISC remain directly associated on polysomes, we investigated the effect of prolonged neuronal hyperactivity on the polysomal distribution of these factors in the context of synaptic homeostasis. To evaluate the hyperactivity-regulated polysome association, density gradient fractions from neurons treated with either bicuculline (10 μM, 24 hours) or vehicle (Figs 5A, 5B, S2H, and S2I) were analyzed by western blot using antibodies against translation regulators, proteasome subunits, chaperone, and members of the miRISC. We observed a relative enrichment of the translation elongation factor eEF2, mTORC1 downstream effector p70 S6K as well as its phosphorylated form, and the phosphorylated ribosomal protein S6 in polysome fractions following prolonged neuronal activity (Fig 5A–5D). Similar to enrichment of translation regulators, we also observed an enrichment of 20S core proteasome subunits upon bicuculline treatment (Fig 5A–5D). The chaperone protein HspA2 was detected in polysome fractions in both vehicle-treated and bicuculline-treated neurons (Fig 5A–5D). The core component of the miRISC, Ago, showed a relative depletion from polysome fractions in bicuculline-treated neurons as compared to vehicle-treated neurons. Furthermore, we observed that the abundance of Trim32 was enhanced (108.3 ± 7.74% increase, $p < 0.005$), whereas MOV10 (72.37 ± 3.54% decrease, $p < 0.002$) was depleted from polysome in response to synaptic hyperactivity (Fig 5C–5F).

Comprehensively, these data point toward activity-induced dynamicity in polysome association of factors regulating synaptic homeostasis.

## Translation drives proteasomal degradation to cause miRISC remodeling during synaptic downscaling

The reciprocal pattern of abundance of MOV10 and Trim32 in polysomes upon chronic hyperactivity (Fig 5E and 5F) and their association with Ago (Fig 4F), a core member of the

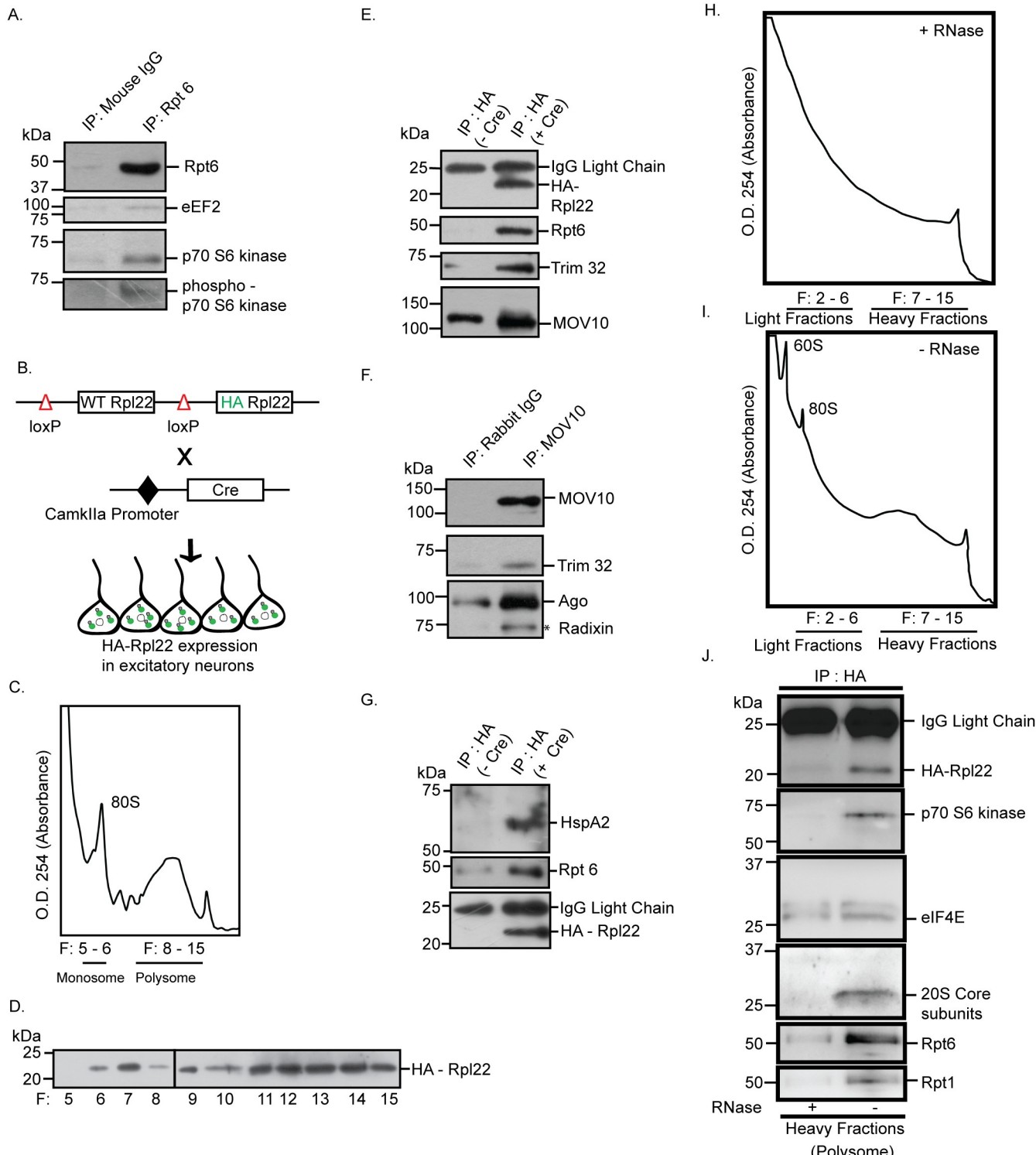

**Fig 4. Interaction between proteasome and actively translating RNA-associated polyribosomes. (A)** Proteasome-associated protein complex was immunoprecipitated from hippocampal lysate using antibody against Rpt6 or mouse IgG. Western blot of purified protein complex performed using antibodies against Rpt6, eEF2, p70 S6K, and phospho-p70 S6K. **(B)** RiboTag mouse when crossed with CamKIIa promoter-driven Cre recombinase mouse results in the deletion of wild-type Rpl22 ribosomal protein and replacement of HA-tagged Rpl22 in forebrain excitatory neurons. **(C)** A$_{254}$ profile showing indicated fractions of monosome and polysome. **(D)** Polysome fractions from (C) showing enrichment of HA-Rpl22 as detected by western blot using antibody against HA. **(E)** HA-tagged Rpl22 containing polyribosome was affinity-purified using antibody against HA. Western blot analysis of affinity-purified complex shows the presence of HA, Rpt6, Trim32, and MOV10. See also S2 Fig. **(F)** MOV10 immunoprecipitated from hippocampal lysates. Western blot analysis of

MOV10-immunoprecipitated protein complex shows the coprecipitation of Trim32 with miRISC components MOV10 and Ago. **(G)** Detection of HspA2 and Rpt6 in HA affinity-purified protein complex from HA-Rpl22 expressing neurons by western blot using antibody against HspA2, Rpt6 and HA. **(H, I)** $A_{254}$ profile showing indicated fractions of monosome and polysome obtained from cytoplasmic lysates treated with (H) or without (I) RNase prior to density gradient fractionation. See also S2 Fig. **(J)** HA-tagged Rpl22 containing ribosomes affinity-purified from heavy fractions of sucrose gradient using antibody against HA. Western blot analysis of affinity-purified complex with antibodies against HA, p70 S6K, eIF4E, 20S Core subunits, Rpt6, and Rpt1. The data underlying this figure are available at https://figshare.com/articles/dataset/Homeostatic_scaling_is_driven_by_a_translation-dependent_degradation_axis_that_recruits_miRISC_remodeling/16768816. Ago, Argonaute; HA, haemagglutinin; IgG, immunoglobulin G; IP, immunoprecipitation; miRISC, miRNA-induced silencing complex; p70 S6K, p70 S6 kinase; WT, wild-type.

miRISC, prompted us to analyze their expression in the context of synaptic downscaling. Bicuculline treatment of hippocampal neurons (DIV 18 to 21) enhanced (108.2 ± 7.55% increase, $p < 0.0001$) Trim32 with a concomitant decrease (65.94 ± 2.67% decrease, $p < 0.001$) in MOV10 (Fig 6A–6C). The increase in Trim32 expression post-bicuculline treatment was blocked by anisomycin and surprisingly resulted in the inhibition of MOV10 degradation (184.82 ± 10.77% protected MOV10, $p < 0.03$) (Fig 6C). This indicates that the degradation of MOV10 is dependent on enhanced Trim32 synthesis and that Trim32 translation precedes the commencement of MOV10 degradation. Treatment with lactacystin resulted in the expected protection of MOV10 from degradation (202.41 ± 3.18% MOV10 protected, $p < 0.006$) upon bicuculline-induced hyperactivity (Fig 6C), whereas there remained no change in the Trim32 expression levels (Fig 6B). Coapplication of lactacystin and anisomycin during bicuculline-induced hyperactivity changed the expression of MOV10 and Trim32 commensurate to basal levels (Fig 6B and 6C). We did not observe any alteration of Ago expression upon prolonged bicuculline treatment of hippocampal neurons (S3A and S3B Fig). Chronic inhibition of protein synthesis alone, without bicuculline treatment, led to a modest but statistically significant decrease of both Trim32 (22.42 ± 0.70% decrease, $p < 0.001$) and MOV10 (28.14 ± 0.48% decrease, $p < 0.0003$) (S3C–S3E Fig), whereas chronic inhibition of the proteasome alone has no effect on Trim32 and MOV10 expression (S3C–S3E Fig) under basal conditions. The significant decrease of both proteins upon anisomycin treatment may be due to the combined effect of global inhibition of translation and the ongoing basal level of protein degradation.

The observed reciprocity between MOV10 and Trim32 expression levels on chronic bicuculline treatment led us to analyze whether Trim32 is the sole E3-ligase responsible for the UPS-mediated degradation of MOV10. Knockdown of Trim32 prevented the bicuculline-induced degradation of MOV10 (199.41 ± 0.69% protected, $p < 0.0001$) (Fig 6D and 6E). Moreover, loss of Trim32 alone enhanced the expression of MOV10 (58.3 ± 3.09% increase, $p < 0.0001$) as compared to basal level (Fig 6D and 6E). We immunoprecipitated MOV10 from bicuculline or vehicle-treated hippocampal neurons expressing shRNA against Trim32 or control shRNA. MOV10 ubiquitination was analyzed by western blot using an antibody that specifically recognizes polyubiquitin conjugates. We observed that bicuculline-induced polyubiquitination of MOV10 was abrogated by Trim32 knockdown (Fig 6F). These observations indicate that (a) the translation of Trim32 is a prerequisite for the degradation of MOV10 by proteasome during synaptic downscaling and (b) Trim32 is the only E3 ligase necessary and sufficient for MOV10 ubiquitination.

Bicuculline-induced changes in Trim32 and MOV10 levels without any change in Ago suggest that during synaptic hyperactivity, miRISC remodeling could occur due to Trim32 translation–dependent MOV10 degradation. To confirm this, we have immunoprecipitated Ago and analyzed the association of key components of the miRISC post-bicuculline treatment for 24 hours. We found a relative depletion of MOV10 and enrichment of Trim32 within the silencing complex upon synaptic hyperactivity (Fig 6G). However, the association of Dicer, another member of the miRISC, remains unaffected (Fig 6G). Comprehensively, our data

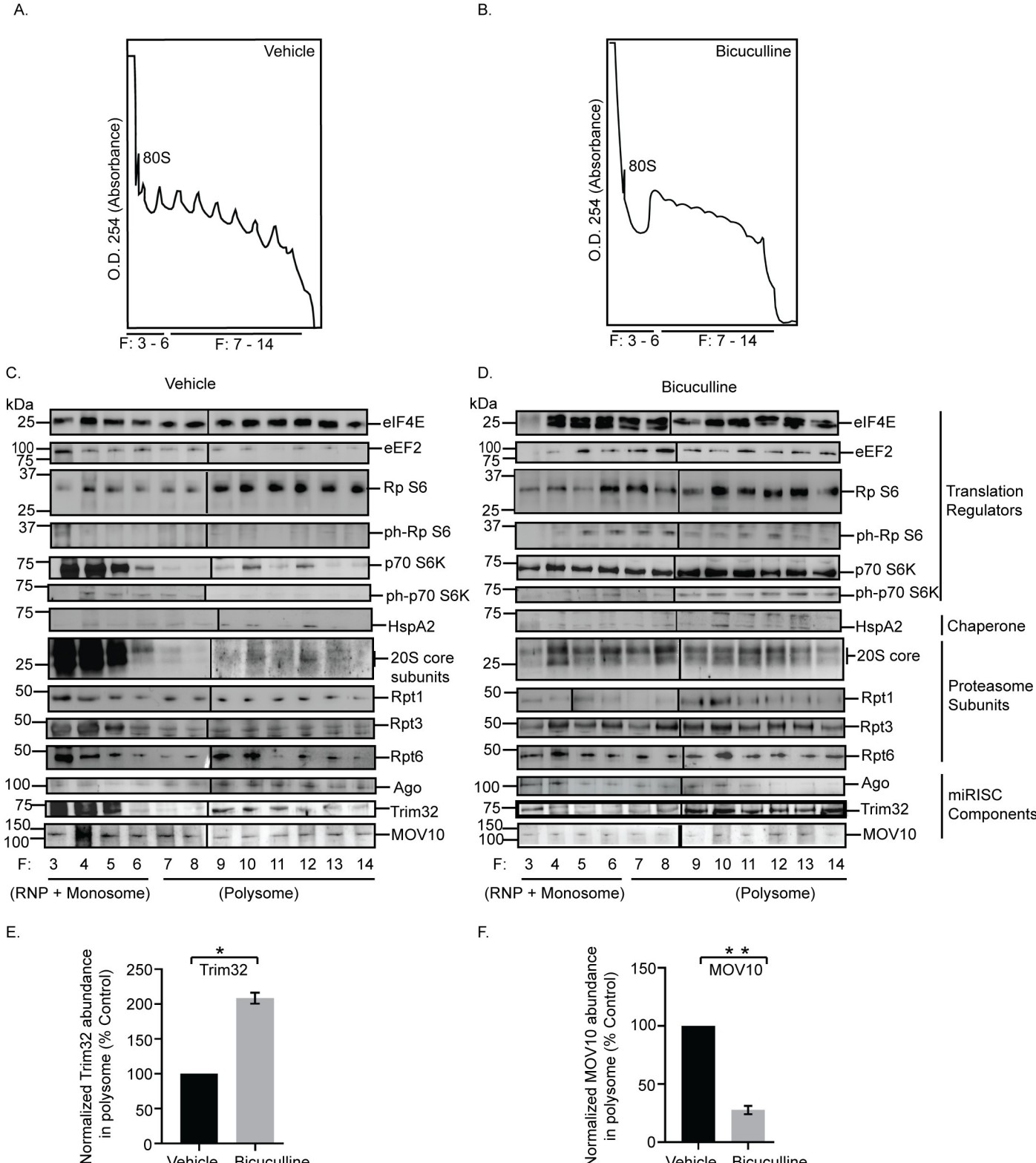

**Fig 5. Hyperactivity-dependent polysome distribution of proteasome, translation regulators, and components of miRISC.** Cortical neurons were treated with bicuculline or vehicle for 24 hours and then subjected to polysome fractionation. Western blot analysis was performed from the tri-chloroacetic acid–precipitated fractions to determine the distribution of translation regulators, miRISC components, chaperone, and the proteasome subunits in the fractions. **(A, B)** $A_{254}$ profile of

fractionated cytoplasmic extracts from cortical neurons treated with vehicle (A) or bicuculine (B). Monosome (80S) or polysome fractions as indicated. **(C, D)** Western blot analysis of fractions from vehicle (C) or bicuculine (D) treated neurons showing the distribution of translation regulators eIF4E, eEF2, Rp S6, ph-Rp S6, p70 S6K, ph-p70 S6K; chaperone protein HspA2; proteasome subunit of 20S core, and Rpt1, Rpt3, Rpt6 of 19S cap, miRISC proteins Ago, MOV10, and Trim32. Rpt1 blots with different exposures are distinguished by a vertical black line to denote that they represent separate panels within the figure. Two blots with different exposures are shown in the main figure and raw data to visualize the specific band of Rpt1. **(E)** Quantitation of polysome distribution of Trim32. **(F)** Quantitation of polysome distribution of MOV10. $^*p < 0.005$, $^{**}p < 0.002$. $n = 3$, Unpaired $t$ test with Welch's correction. See also S2 Fig. The data underlying this figure are available at https://figshare.com/articles/dataset/Homeostatic_scaling_is_driven_by_a_translation-dependent_degradation_axis_that_recruits_miRISC_remodeling/16768816. Ago, Argonaute; miRISC, miRNA-induced silencing complex; ph-Rp S6, phosphorylated ribosomal protein S6; ph-p70 S6K, phosphorylated p70 S6 kinase; p70 S6K, p70 S6 kinase; RNP, ribonucleoprotein; Rp S6, ribosomal protein S6.

point toward chronic network hyperactivity-induced miRISC remodeling that alters the expression of the miRISC members, Trim32 and MOV10.

## Translation of Trim32 during chronic hyperactivity is mTORC1 dependent

Coprecipitation of the downstream effectors of the mTORC1 signaling cascade with the 26S proteasomal subunit Rpt6 (Fig 4A) and the activity-dependent differential distribution of these effectors in polysome (Fig 5C and 5D) led us to examine whether mTORC1 signaling plays a role in causing synaptic downscaling in response to chronic hyperactivity. Bicuculline treatment of hippocampal neurons in the presence of rapamycin (100 nM, 24 hours), a selective inhibitor of mTORC1, completely abolished the chronic hyperactivity-driven Trim32 synthesis (16.48 ± 10.33% increase as compared to control, $p = 0.45$) and consecutive MOV10 degradation (8.19 ± 2.44% decrease as compared to control, $p = 0.06$) (Fig 7A–7C). Rapamycin treatment alone did not alter the expression patterns of Trim32 and MOV10 (Fig 7A–7C). This led us to hypothesize that mTORC1 pathway acts upstream of Trim32, serving to regulate its synthesis in response to chronic bicuculline treatment. Consistent with our biochemical data, we observed that coincubation of rapamycin and bicuculline prevented the decrease in mEPSC amplitude (2.76 ± 0.13 pA increase as compared to bicuculline-treated neurons, $p < 0.01$) but not frequency (Fig 7D–7F). Just as above, rapamycin treatment alone has no effect, indicating that chronic hyperactivity acts as a triggering point for mTORC1 activation (Fig 7D–7F) and this subsequently plays a role in driving Trim32 translation.

mTORC1-mediated translation is regulated by the phosphorylation of two divergent downstream effectors, eukaryotic translation initiation factor 4E-binding protein 2 (4E-BP2) and p70 S6K [37]. We examined the phosphorylation status of these two downstream effectors following bicuculline-induced hyperactivity. Chronic bicuculline treatment leads to a significant enhancement of p70 S6K phosphorylation (74.66 ± 2.61% increase, $p < 0.0001$), which was blocked by rapamycin (Fig 8A and 8B). However, bicuculline treatment did not affect the phosphorylation of 4E-BP2 (Fig 8A and 8C).

We used a specific inhibitor of p70 S6K, LY2584702 Tosylate (2 μM, 24 hours), in primary hippocampal neurons to evaluate the role of p70 S6K activity on Trim32 and MOV10 expression in the context of synaptic downscaling. Similar to rapamycin treatment, the inhibition of p70 S6K phosphorylation prevented Trim32 translation and the subsequent degradation of MOV10 upon bicuculline-induced hyperactivity (Fig 8D–8F). These observations showed that the activation of p70 S6K by mTORC1 is a key determinant regulating the concerted translation of Trim32 and the degradation of MOV10 in downscaling.

## MOV10 degradation is sufficient to invoke downscaling of AMPARs

MOV10 degradation in response to chronic bicuculline treatment made us enquire whether its loss alone was sufficient to cause pervasive changes in the miRISC to effectuate synaptic downscaling. We used lentivirus-mediated RNAi of MOV10 to mimic hyperactivity-driven

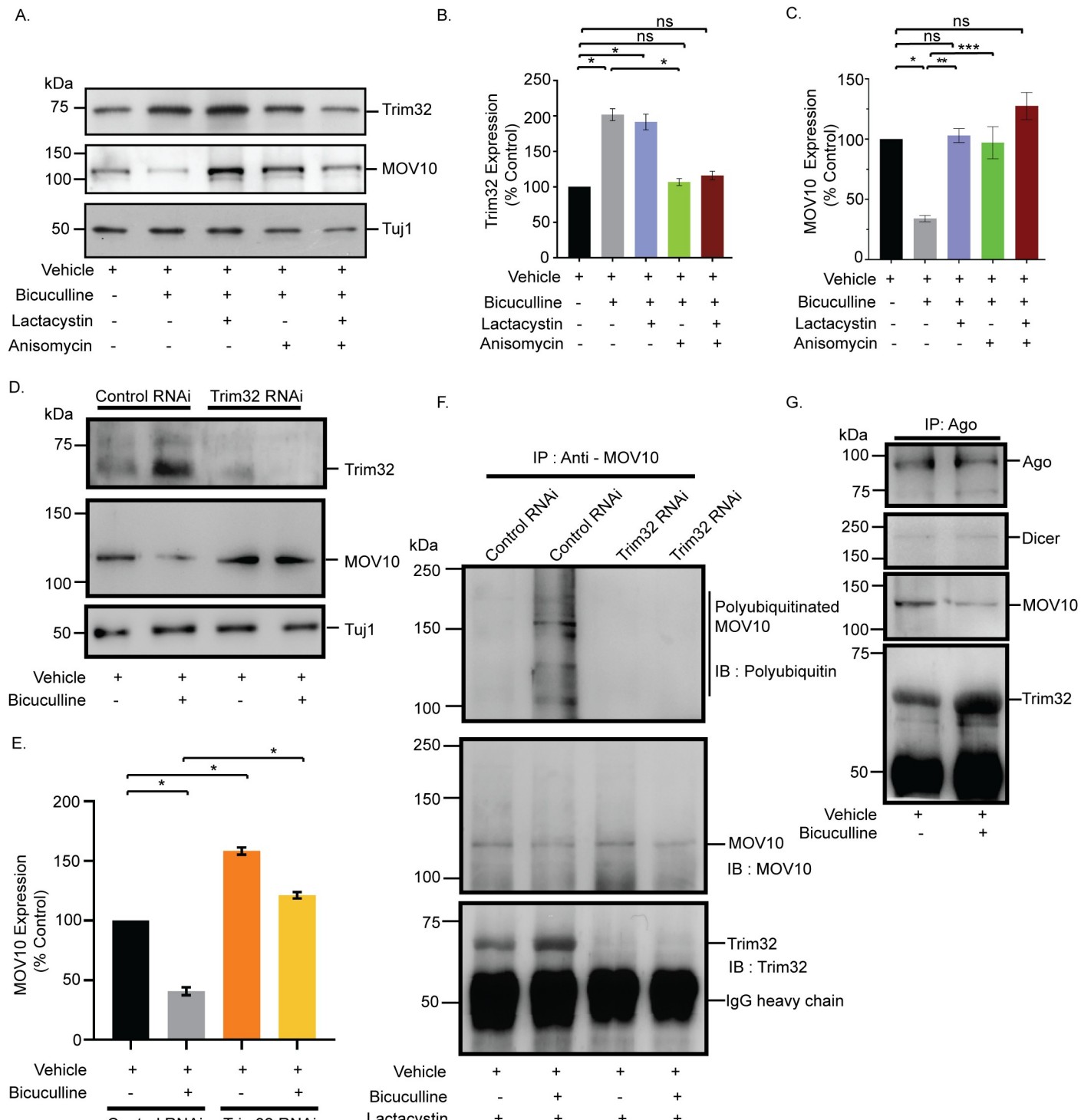

**Fig 6. Synthesis of Trim32 facilitates polyubiquitination and subsequent degradation of MOV10 for miRISC remodeling. (A)** Western blot analysis showing the expression of Trim32, MOV10, and Tuj1 from neurons treated with bicuculline with or without lactacystin, anisomycin or both. **(B)** Quantitation of Trim32 expression. **(C)** Quantitation of MOV10 expression. $n$ = 3. Data shown as mean ± SEM. $^*p < 0.0001$ (B) and $^*p < 0.001$, $^{**}p < 0.006$, $^{***}p < 0.03$ (C). ns, not significant. See also S3 Fig. One-way ANOVA and Fisher's LSD. **(D)** Western blot analysis of Trim32 RNAi neurons or control RNAi neurons treated with bicuculline or vehicle, showing the expression of Trim32, MOV10, and Tuj1. **(E)** Quantitation of MOV10. $n$ = 5. Data shown as mean ± SEM. $^*p < 0.0001$. Unpaired $t$ test with Welch's correction. **(F)** IP of MOV10 from hippocampal neurons treated with bicuculline or vehicle in presence of lactacystin. Polyubiquitination of MOV10 detected by western blot analysis using an antibody against polyubiquitin conjugates (FK1). Western blot analysis of MOV10-immunoprecipitated protein complex shows the coprecipitation of Trim32 in presence or absence of bicuculline. **(G)** Ago immunoprecipitated from hippocampal neurons treated with

bicuculline or vehicle. Western blot analysis of immunoprecipitated Ago complex shows the coprecipitation of the miRISC proteins Dicer, MOV10, and Trim32. The data underlying this figure are available at https://figshare.com/articles/dataset/Homeostatic_scaling_is_driven_by_a_translation-dependent_degradation_axis_that_recruits_miRISC_remodeling/16768816. Ago, Argonaute; IB, immunoblot; IgG, immunoglobulin G; IP, immunoprecipitation; miRISC, miRNA-induced silencing complex; ns, not significant.

MOV10 degradation. Intensity of sGluA1/A2 puncta that colocalized with PSD95 was analyzed following MOV10 knockdown (DIV 21 to 24). We observed that loss of MOV10 reduced the expression of sGluA1 (35.03 ± 9.35% for shRNA#1, $p < 0.01$ and 58.38 ± 10.27% for shRNA#2, $p < 0.01$) and sGluA2 (49.4 ± 12.9% for shRNA#1, $p < 0.01$) at the synapses (Figs 9A–9D and S4), which recapitulated the redistribution of sGluA1/sGluA2 in neurons under

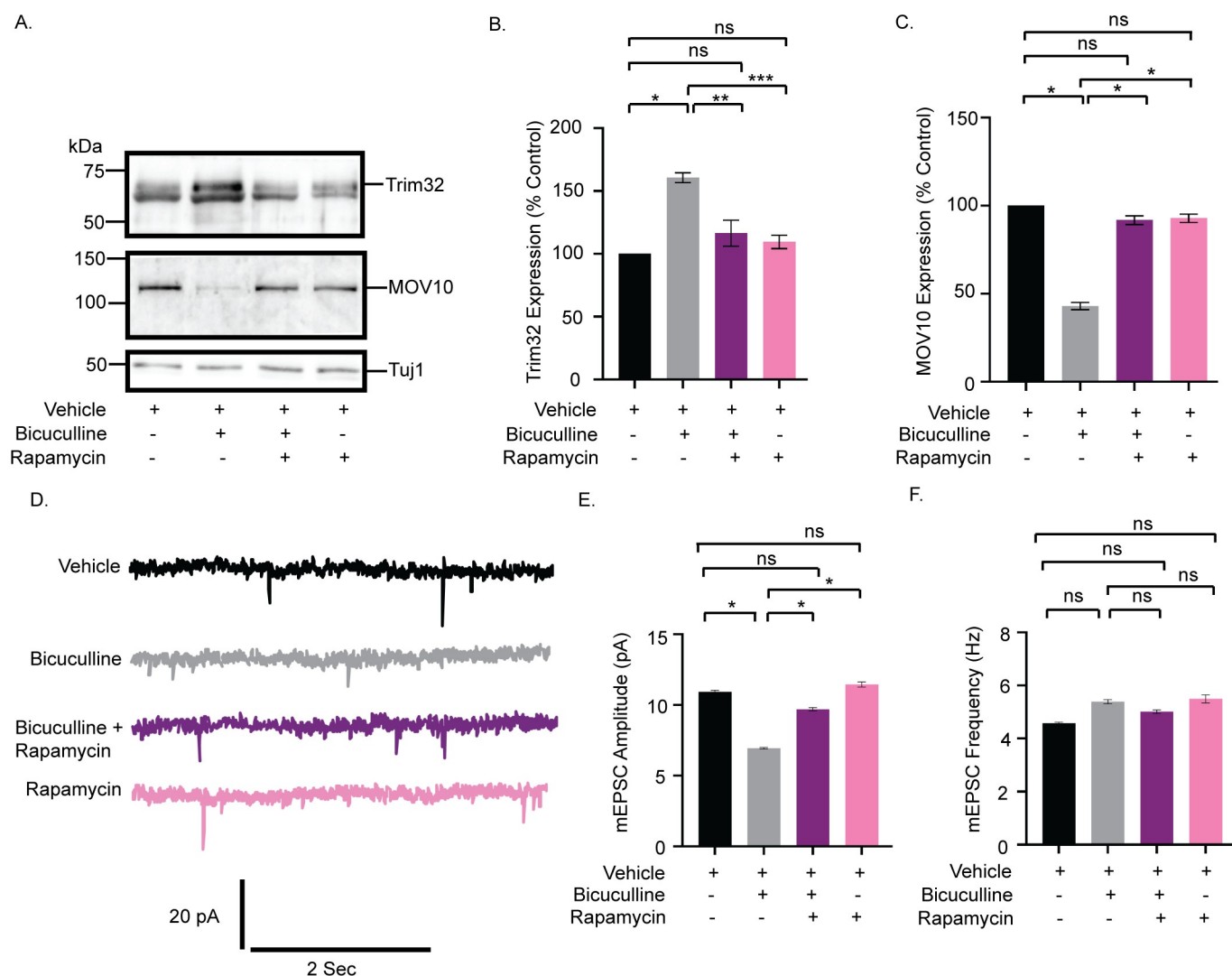

**Fig 7. mTORC1 drives Trim32 synthesis and consequent MOV10 degradation during synaptic downscaling. (A)** Western blot analysis from neurons treated with bicuculline, rapamycin, or both showing the expression levels of Trim32, MOV10, and Tuj1. **(B)** Quantitation of Trim32 expression. **(C)** Quantitation of MOV10 expression. Data shown as mean ± SEM. $n = 5$. $^*p < 0.0001$, $^{**}p < 0.0007$, $^{***}p < 0.0001$ (B) and $^*p < 0.001$ (C). ns, not significant. One-way ANOVA and Bonferroni's correction. **(D)** mEPSC traces from neurons treated with vehicle, bicuculline, rapamycin, or both. **(E)** Mean mEPSC amplitude. **(F)** Mean mEPSC frequency. $n = 8$–9. $^*p < 0.01$. ns, not significant. Data shown as mean ± SEM. One-way ANOVA and Fisher's LSD. The data underlying this figure are available at https://figshare.com/articles/dataset/Homeostatic_scaling_is_driven_by_a_translation-dependent_degradation_axis_that_recruits_miRISC_remodeling/16768816. mEPSC, miniature excitatory postsynaptic current; mTORC1, mammalian Target Of Rapamycin Complex-1; ns, not significant.

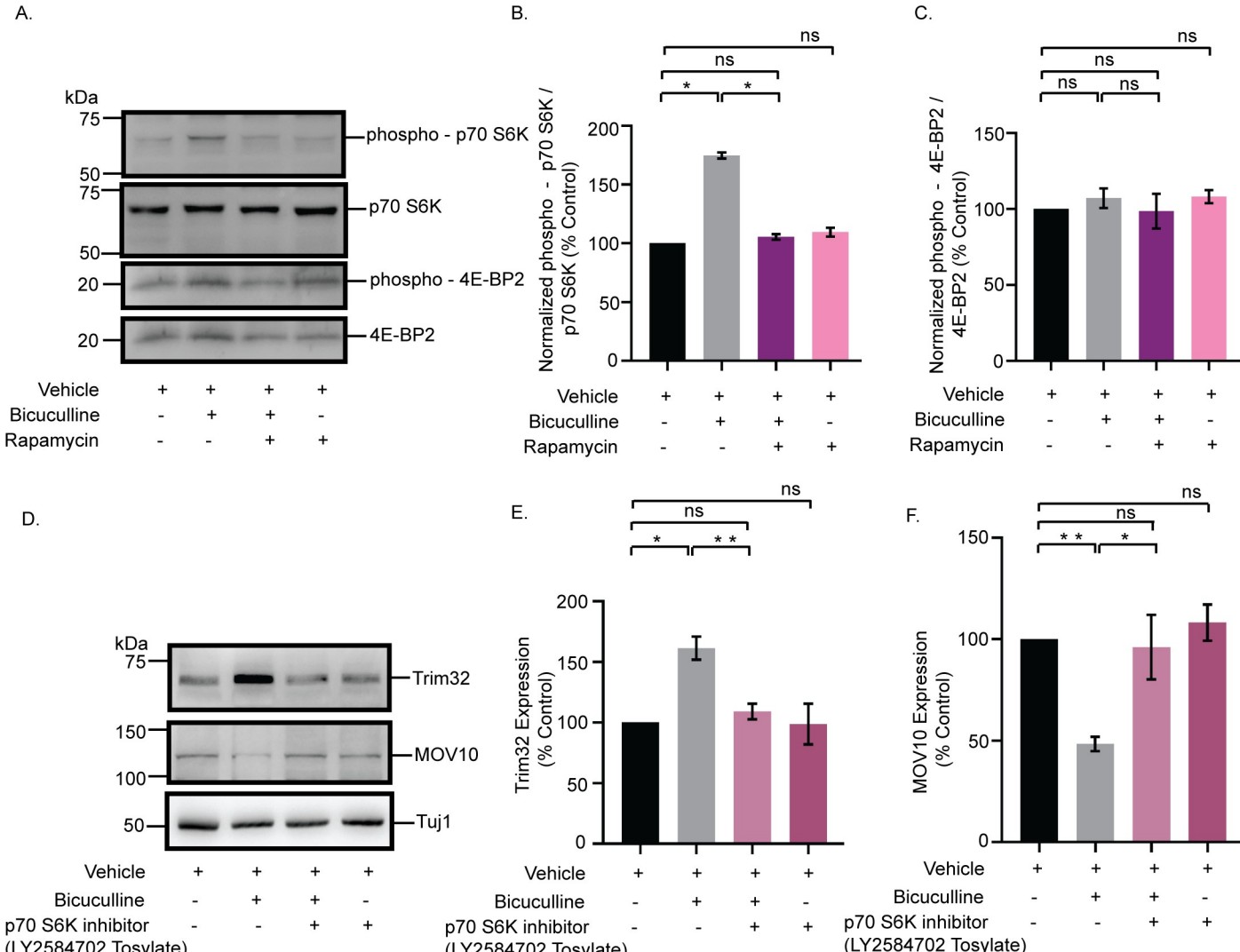

**Fig 8. mTORC1 regulates the Trim32-MOV10 axis *via* the phosphorylation of p70 S6K. (A)** Western blot analysis from neurons treated with bicuculline, rapamycin, or both showing the phosphorylation and total expression of p70 S6K and 4E-BP2. **(B)** Quantitation of p70 S6K phosphorylation. **(C)** Quantitation of 4E-BP2 phosphorylation. $n = 4$. $*p < 0.0001$. ns, not significant. Data shown as mean ± SEM. One-way ANOVA and Fisher's LSD. **(D)** Western blot analysis from neurons treated with bicuculline, p70 S6K inhibitor LY2584702 Tosylate, or both, showing the expression of MOV10 and Trim32. **(E)** Quantitation of Trim32 expression. **(F)** Quantitation of MOV10 expression. $n = 5$. $*p < 0.003$, $**p < 0.0002$ (E) $**p < 0.0001$, $*p < 0.02$ (F). ns, not significant. Data shown as mean ± SEM. One-way ANOVA and Fisher's LSD. The data underlying this figure are available at https://figshare.com/articles/dataset/Homeostatic_scaling_is_driven_by_a_translation-dependent_degradation_axis_that_recruits_miRISC_remodeling/16768816. mTORC1, mammalian Target Of Rapamycin Complex-1; ns, not significant; p70 S6K, p70 S6 kinase; 4E-BP2, 4E-binding protein 2.

chronic bicuculline treatment (Fig 2C and 2D). The knockdown of MOV10 reduced mEPSC amplitude (3.43 ± 0.16 pA for shRNA#1 and 4.35 ± 0.14 pA for shRNA#2, $p < 0.01$) but not frequency (Fig 9E–9G), an observation that mirrors bicuculline-induced synaptic downscaling (Fig 1E).

We examined whether MOV10 knockdown could further influence synaptic strength effectuated by bicuculline. Bicuculline treatment did not further decrease the amplitude nor change the frequency of mEPSCs recorded from neurons expressing MOV10 shRNAs (Fig 10A–10C). This observation indicates that effect of MOV10 knockdown overrides the effect of bicuculline. We then overexpressed myc-tagged MOV10 in hippocampal neurons and measured their

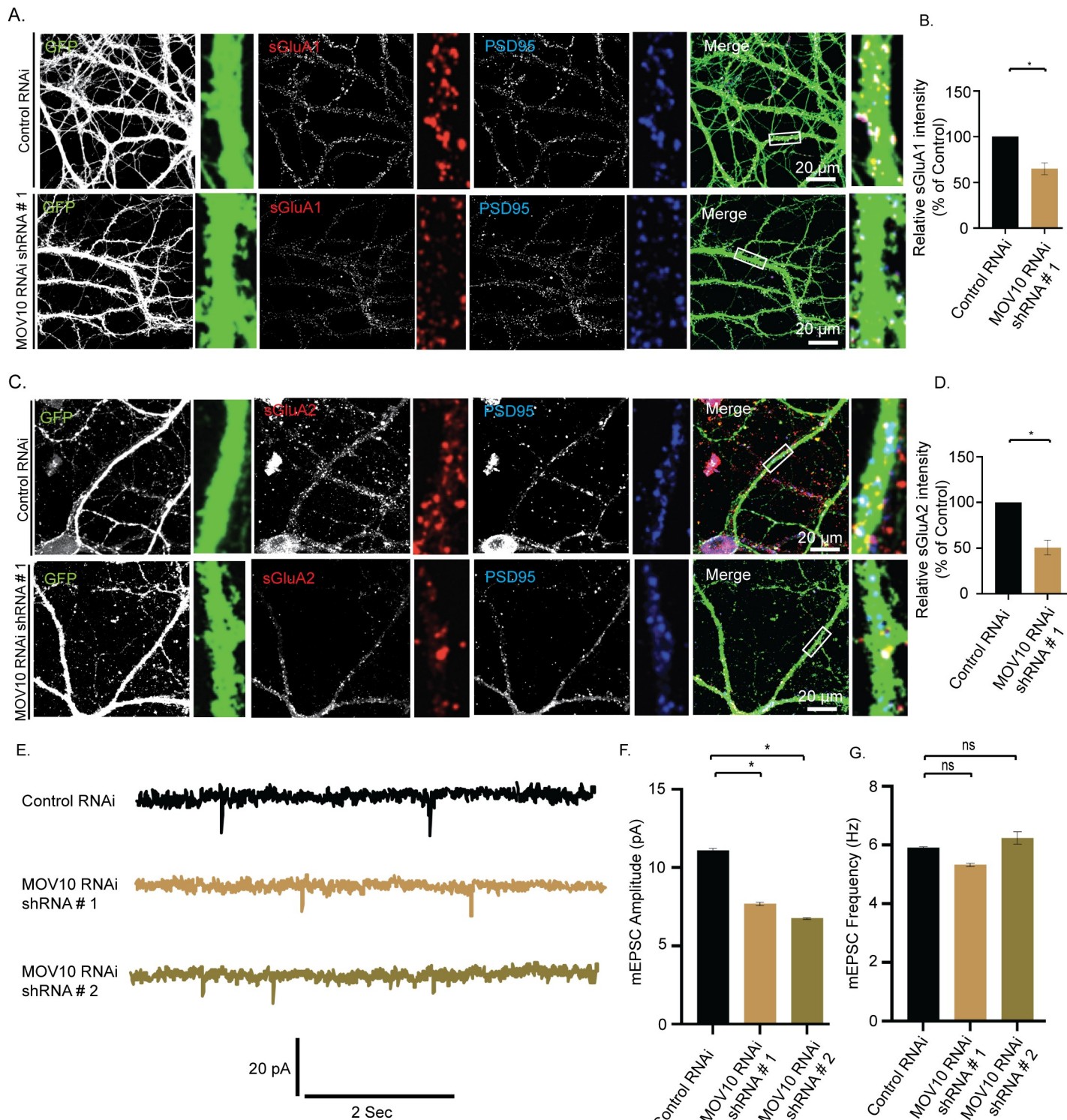

**Fig 9. MOV10 modulates the abundance of sAMPARs during synaptic downscaling.** High-magnification images of neurons transduced with lentivirus coexpressing EGFP and shRNA against MOV10 (MOV10 RNAi) or nontargeting shRNA (Control RNAi) showing the expression of sGluA1 or sGluA2. **(A)** Control RNAi and MOV10 RNAi neurons showing sGluA1 (red), PSD95 (blue), GFP (green), and GFP/sGluA1/PSD95 (merged). **(B)** Quantitation of normalized intensity of synaptic sGluA1. **(C)** Control RNAi and MOV10 RNAi neurons showing sGluA2 (red), PSD95 (blue), GFP (green), and GFP/sGluA2/PSD95 (merged). **(D)** Quantitation of normalized intensity of synaptic sGluA2. $n$ = 26–30, GluA1; $n$ = 12–15, GluA2. Data shown as mean ± SEM. $^{*}p$ < 0.01. One-way ANOVA and Fisher's LSD. Dendrite marked in white box was digitally amplified. See also S4 Fig. **(E)** mEPSC traces from neurons transduced with shRNAs against MOV10 or control shRNA. **(F)** Mean mEPSC amplitude. **(G)** Mean mEPSC frequency. $n$ = 12–13. $^{*}p$ < 0.01. ns, not significant. Data shown as mean ± SEM. One-way ANOVA and Fisher's LSD. The data

underlying this figure are available at https://figshare.com/articles/dataset/Homeostatic_scaling_is_driven_by_a_translation-dependent_degradation_axis_that_recruits_miRISC_remodeling/16768816. mEPSC, miniature excitatory postsynaptic current; ns, not significant; sAMPAR, surface AMPAR.

synaptic activity. Bicuculline treatment led to an enhancement of mEPSC amplitude ($1.64 \pm 0.14$ pA increase, $p < 0.0001$) (Fig 10D and 10E) in neurons overexpressing MOV10 but not mEPSC frequency (Fig 10D and 10F). Our data show that the overexpression of MOV10 partially occludes synaptic downscaling.

## Trim32 is required for AMPAR-mediated downscaling

We have examined the role of Trim32 in bicuculline-induced downscaling by measuring synaptic activity following the loss of Trim32. Trim32 knockdown in bicuculline-treated neurons led to an increase in mEPSC amplitude ($2.82 \pm 0.23$ pA increase, $p < 0.0001$) but did not show any change in frequency as compared to neurons incubated with bicuculline alone (Fig 11A–11C). A modest but significant increase in mEPSC amplitude ($1.65 \pm 0.25$ pA increase, $p < 0.0001$) was also detected from neurons expressing Trim32 shRNA as compared to control shRNA without any activity (Fig 11A and 11B). We presume that this could be due to an increase in basal MOV10 expression following Trim32 knockdown (Fig 6D and 6E). Consistent with patch clamp recording data from MOV10 overexpressing neurons, a partial occlusion of bicuculline-induced downscaling by Trim32 knockdown establishes the requirement of Trim32-MOV10 axis in synaptic scaling. We examined the involvement of sAMPARs in regulation of Trim32-mediated downscaling. We observed that Trim32 knockdown in bicuculline-treated neurons enhanced both sGluA1 ($31.35 \pm 6.65\%$ increase, $p < 0.03$) (Fig 12A, 12B, and 12E) and sGluA2 ($88.99 \pm 5.52\%$ increase, $p < 0.002$) (Fig 12C, 12D, and 12F) as compared to neurons treated with bicuculline alone.

Similar to increase in mEPSC amplitude following Trim32 knockdown, our data showed an increase in sGluA1 ($27.009 \pm 7.72\%$, $p < 0.02$) and sGluA2 ($61.63 \pm 10.47\%$, $p < 0.0004$) in Trim32 RNAi neurons (Fig 12E and 12F). Comprehensively, our data demonstrate a regulatory control of synaptic downscaling by the Trim32-MOV10 axis that modulates sAMPAR expression.

## The Trim32-MOV10 axis regulates Arc expression during synaptic downscaling

How does MOV10 degradation lead to the removal of sAMPARs to regulate synaptic downscaling? Arc/Arg3.1, an immediate early gene, has been shown to be dynamically regulated by chronic changes in synaptic activity and evokes synaptic scaling [38]. Overexpression of Arc decreases sAMPARs *via* endocytosis, whereas its knockdown increases them [39]. Arc expression has been shown to be regulated by diverse mechanisms including translational control that involves miRNAs [40–42]. Since we found a decrease of sAMPAR distribution on MOV10 knockdown, we investigated the correlation between MOV10 and Arc expression. We observed that the loss of MOV10 enhanced Arc ($113.1 \pm 15.7\%$ increase, $p < 0.002$ for shRNA #1 and $173.8 \pm 7.45\%$ increase, $p < 0.0001$ for shRNA #2) (Fig 13A and 13B). The extent of increase in Arc protein was commensurate with the degree of MOV10 knockdown (Fig 13A and 13B). We also observed that this differential enhancement of Arc protein was reflected in the proportionate removal of sAMPARs and the concomitant decrease in mEPSC amplitude (Fig 9F). Bicuculline-induced chronic hyperactivity, which degrades MOV10, also enhanced Arc expression ($132.1 \pm 27.45\%$ increase, $p < 0.04$) (Fig 13C and 13D). This activity-driven increase was blocked by the inhibition of mTORC1 with rapamycin (100 nM, 24 hours), whereas rapamycin treatment alone had no effect (Fig 13C and 13D).

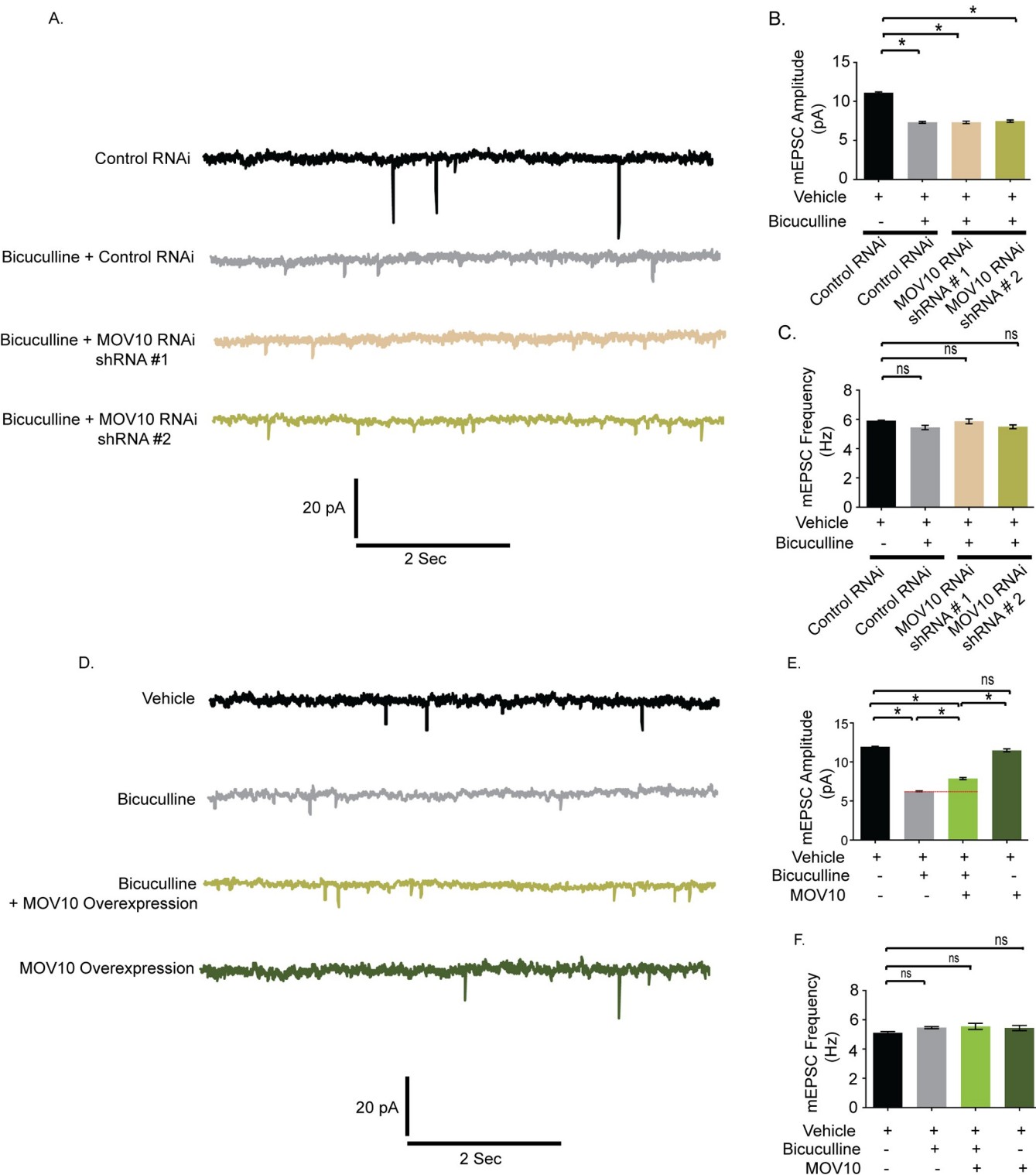

**Fig 10. Overexpression of MOV10 occludes bicuculline–induced synaptic downscaling.** (**A**) mEPSC traces from bicuculline- or vehicle-treated neurons transduced with shRNAs against MOV10 or control shRNA. (**B**) Mean mEPSC amplitude. (**C**) Mean mEPSC frequency. $n = 12–18$. $^*p < 0.0001$. ns, not significant. Data shown as mean ± SEM. One-way ANOVA and Fisher's LSD. (**D**) mEPSC traces from bicuculline-treated neurons overexpressing myc-tagged MOV10. (**E**) Mean mEPSC amplitude. (**F**) Mean mEPSC frequency. $n = 13–15$. $^*p < 0.0001$. ns, not significant. Data shown as mean ± SEM. One-way ANOVA and Fisher's LSD. Dotted line indicates the difference in mEPSC amplitude recorded from neurons overexpressing MOV10 or control neurons

following bicuculline treatment. See also S6 Fig. The data underlying this figure are available at https://figshare.com/articles/dataset/Homeostatic_scaling_is_driven_by_a_translation-dependent_degradation_axis_that_recruits_miRISC_remodeling/16768816.mEPSC, miniature excitatory postsynaptic current; ns, not significant.

Arc expression has been shown to be regulated by both transcriptional and posttranscriptional control [42]. Posttranscriptional control of Arc expression is regulated by its 3′ UTR that contains multiple miRNA-binding sites [40]. We explored the mechanism of mTORC1-dependent Arc expression involving the Trim32-MOV10 axis. We used an Arc-3′ UTR fused luciferase reporter (Arc-Luc) to assess Arc expression from bicuculline-treated neurons following Trim32 and MOV10 knockdown. We found that the bicuculline-induced enhancement of reporter activity (61.1 ± 3.24% increase, $p < 0.0003$) was prevented by Trim32 knockdown (Fig 13E), whereas MOV10 knockdown alone was sufficient to increase reporter activity under basal conditions (77.34 ± 7.78% increase, $p < 0.002$). Loss of MOV10 in bicuculline-treated neurons did not enhance Arc-Luc reporter activity further as compared to bicuculline-treated neurons (Fig 13E). This observation is consistent with our electrophysiology data demonstrating that MOV10 knockdown in bicuculline-treated neurons did not elicit further reduction in mEPSC amplitude when compared to neurons treated with bicuculline alone (Fig 10B).

Since Dicer is a key component of the miRISC, its knockdown would definitely result in pervasive miRISC remodeling and allow us to confirm whether miRISC remodeling has a role in regulating Arc expression. Hence, we tested Arc-Luc reporter activity following Dicer

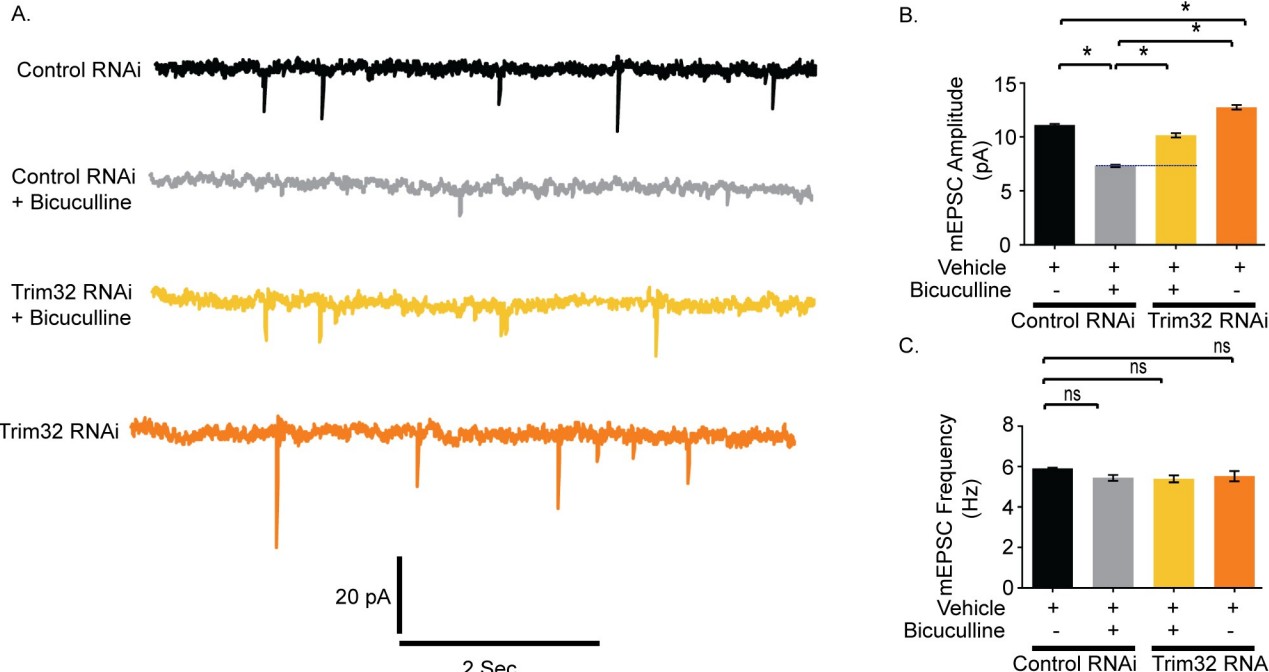

**Fig 11. Trim32 knockdown prevents bicuculline–induced synaptic downscaling. (A)** mEPSC traces from bicuculline- or vehicle-treated neurons transduced with shRNAs against Trim32 or control shRNA. **(B)** Mean mEPSC amplitude. **(C)** Mean mEPSC frequency. $n$ = 12–16. $^*p < 0.0001$. ns, not significant. Data shown as mean ± SEM. One-way ANOVA and Fisher's LSD. Dotted line indicates the difference in mEPSC amplitude recorded from bicuculline-treated neurons transduced with Trim32 shRNA or control shRNA. The data underlying this figure are available at https://figshare.com/articles/dataset/Homeostatic_scaling_is_driven_by_a_translation-dependent_degradation_axis_that_recruits_miRISC_remodeling/16768816. mEPSC, miniature excitatory postsynaptic current; ns, not significant.

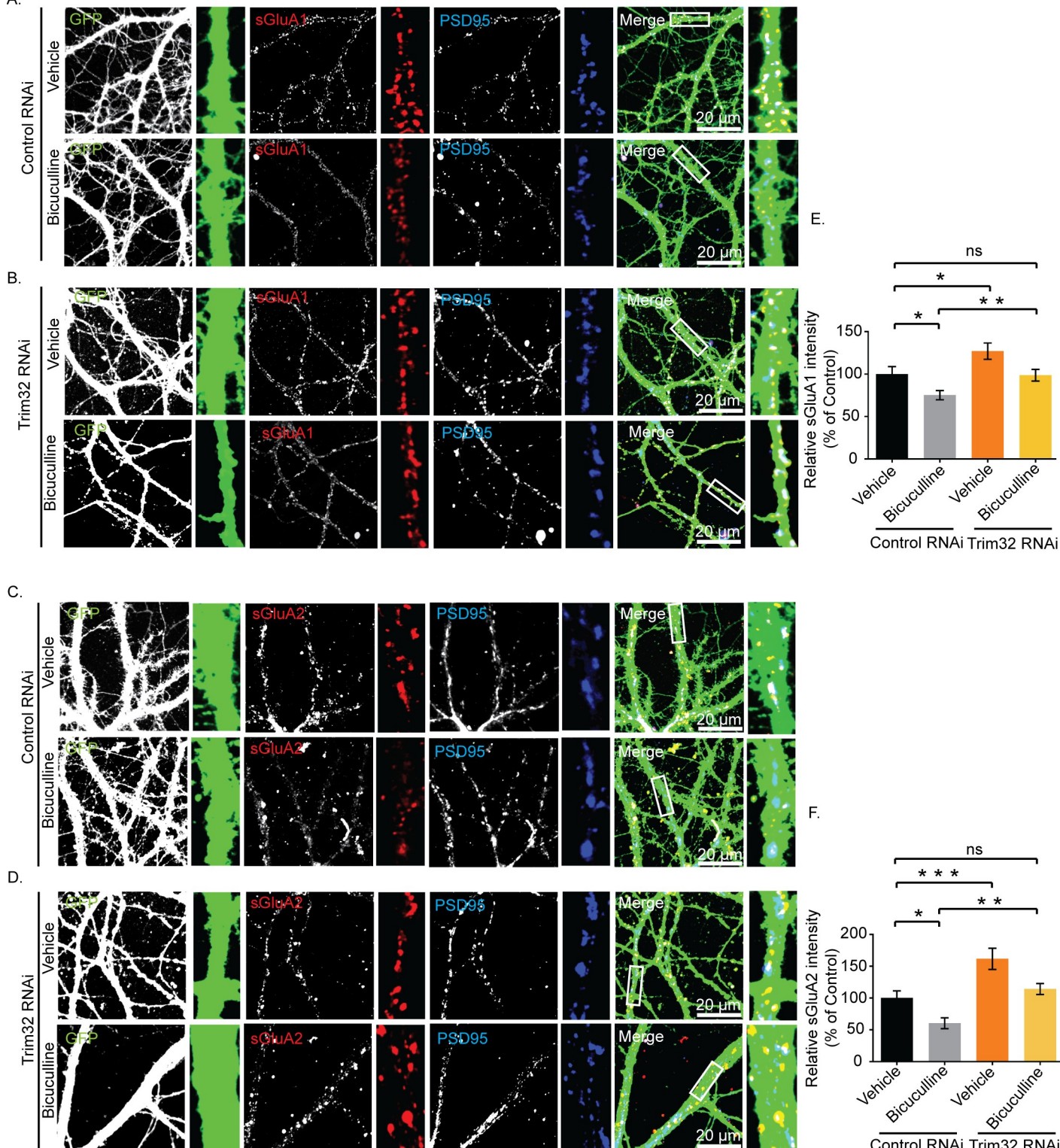

**Fig 12. Trim32-mediated synaptic downscaling occurs *via* modulation of sAMPARs.** Neurons were transduced with lentivirus coexpressing EGFP and shRNA against Trim32 or a nontargeting shRNA and the expression of sGluA1 and sGluA2 determined by immunostaining. **(A, B)** High-magnification images of bicuculline- and vehicle-treated neurons expressing nontargeting shRNA (Control RNAi) (A) or expressing shRNA against Trim32 (Trim32 RNAi) (B) showing the expression of sGluA1 (red), PSD95 (blue), GFP (green), and GFP/sGluA1/PSD95 (merged). **(C, D)** High-magnification images of bicuculline- and vehicle-treated neurons expressing nontargeting shRNA (Control RNAi) (C) or shRNA against Trim32 (Trim32 RNAi) (D) showing the expression of sGluA2 (red), PSD95 (blue),

GFP (green), and GFP/sGluA2/PSD95 (merged). **(E)** Quantitation of normalized intensity of synaptic sGluA1. **(F)** Quantitation of normalized intensity of synaptic sGluA2. $n = 30–40$ for GluA1; $n = 23–36$ for GluA2. Data shown as mean ± SEM. $^*p < 0.02$, $^{**}p < 0.03$, for sGluA1. $^*p < 0.008$, $^{**}p < 0.002$, $^{***}p < 0.0004$ for sGluA2. One-way ANOVA and Fisher's LSD. Dendrite marked in white box was digitally amplified. See also S5 Fig. The data underlying this figure are available at https://figshare.com/articles/dataset/Homeostatic_scaling_is_driven_by_a_translation-dependent_degradation_axis_that_recruits_miRISC_remodeling/16768816. ns, not significant; sAMPAR, surface AMPAR.

knockdown and found that similar to loss of MOV10, Dicer knockdown alone was sufficient to enhance Arc-Luc reporter activity under basal conditions ($77.4 ± 8.31\%$ increase, $p < 0.002$) (Fig 13F), and this enhancement was comparable to bicuculline-induced reporter activity ($77.4 ± 8.31\%$ increase in Dicer-knockdown neurons versus $61.1 ± 3.24\%$ increase in bicuculline-treated neurons) (Fig 13F). Similar to our western blot data (Fig 13C and 13D), we observed that the bicuculline-induced enhancement of Arc-Luc activity ($87.6 ± 11.64\%$ increase, $p < 0.002$) was prevented by the application of rapamycin. Application of rapamycin alone did not show any change (Fig 13G).

To explore whether Arc was regulated by transcription during synaptic downscaling, we checked Arc mRNA expression in bicuculline-treated neurons after MOV10 knockdown and after mTORC1 inhibition. qRT-PCR analysis showed no detectable change in Arc transcript levels upon MOV10 knockdown or bicuculline-induced hyperactivity in presence or absence of rapamycin (Fig 13H and 13I).

Taken together, our data demonstrate that the bicuculline-induced downscaling of synaptic strength occurs via an mTORC1-mediated Trim32 translation-dependent MOV10 degradation involving removal of sAMPARs *via* Arc (Fig 14).

## Discussion

Here, we provide empirical evidence emphasizing that synchrony between protein synthesis and proteasomal activity is critical to establish homeostasis at synapses. We used a paradigm of chronic network hyperactivity to invoke downscaling and determined that (a) translation and degradation apparatuses directly interact with each other and are tethered together by RNA scaffolds; (b) it is the translation of Trim32 that drives the degradation of MOV10 to cause miRISC remodeling, thus the current paradigm is an example of translation preceding degradation; and (c) miRISC is a key node in the translation-degradation axis, with the mTORC1-p70 S6K pathway being the upstream signaling component and a part of the "sensor" machinery, and Arc-induced removal of sAMPARs being the final effectors of downscaling.

### Coregulation of protein synthesis and degradation drives AMPAR-mediated synaptic downscaling

We find that chronic perturbation of either translation or proteasomal activity occludes synaptic homeostasis, whereas homeostasis remains unperturbed when there is simultaneous inhibition of both (Fig 1). Chronic application of bicuculline along with either lactacystin or anisomycin leads to alterations of mEPSC amplitude that exactly mirror observations where bicuculline is absent (Fig 1B versus Fig 1E). Thus, the effects of bicuculline-induced changes to the existing proteome are overshadowed by those accomplished by the individual action of the proteasome or the translation machinery (Fig 1). The importance of these observations is multifaceted; it establishes that (i) congruent protein synthesis and degradation pathways regulate synaptic scaling; (ii) the constancy of the proteomic pool in the presence of lactacystin and anisomycin renders the effect of any network destabilizing stimuli like bicuculline to be redundant; and (iii) bicuculline-induced changes in the proteome predominantly affect the physiology of the postsynaptic compartment.

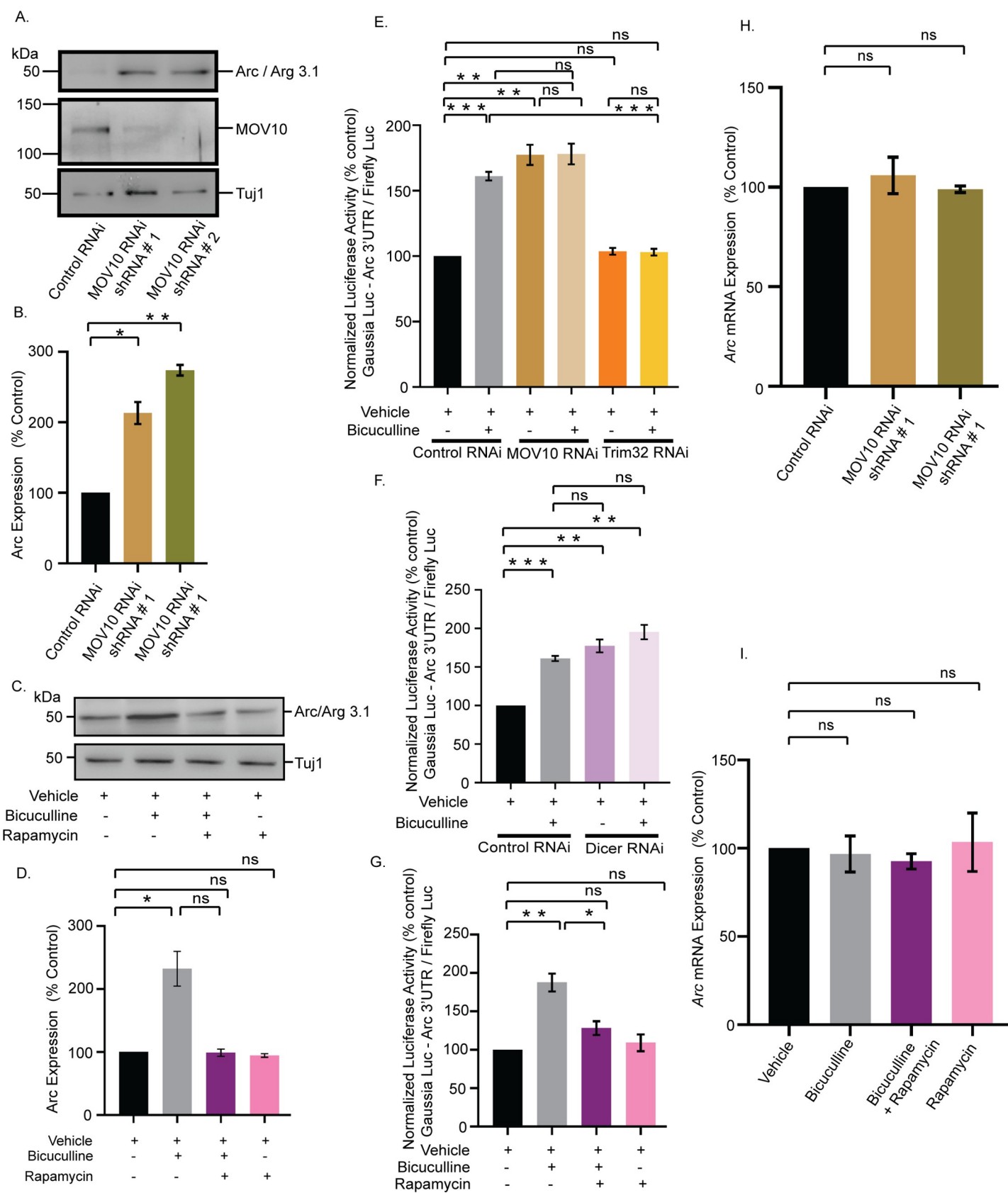

**Fig 13. mTORC1-mediated regulation of Arc expression upon chronic hyperactivity involves MOV10. (A)** Western blot analysis showing the Arc protein level after MOV10 knockdown in neurons infected with lentivirus expressing two different shRNAs against MOV10. **(B)** Quantitation of Arc expression. $n = 6$. $^*p < 0.002$, $^{**}p < 0.0001$. Data shown as mean ± SEM. One-way ANOVA and Fisher's LSD. **(C)** Western blot analysis of neurons treated with bicuculline in presence or absence of rapamycin showing the expression of Arc protein. **(D)** Quantitation of Arc expression. $n = 3$. $^*p < 0.04$. Data shown as mean ± SEM. One-way ANOVA and Fisher's LSD. **(E, F)** Quantitation of luciferase reporter expression from vehicle- or bicuculline-treated neurons transduced with lentivirus expressing shRNA against MOV10 and Trim32 (E), and Dicer (F). $n = 4$. $^{**}p < 0.002$, $^{***}p < 0.0003$, ns, not significant. Data shown as mean ± SEM. One-way ANOVA and Fisher's LSD. See also S6 Fig. **(G)** Quantitation of luciferase reporter expression from vehicle- or bicuculline-treated neurons in presence or absence of rapamycin. $n = 7$. $^{**}p < 0.002$, $^*p < 0.04$, ns, not significant. Data shown as mean ± SEM. One-way ANOVA and Fisher's LSD. **(H)** Quantitation of qPCR analysis of Arc mRNA expression in neurons expressing shRNA against MOV10 or control shRNA. $n = 3$. ns, not significant. Data shown as mean ± SEM. One-way ANOVA. **(I)** Quantitation of qPCR analysis of Arc mRNA expression in bicuculline-treated neurons in presence or absence of rapamycin. $n = 4$. ns, not significant. Data shown as mean ± SEM. One-way ANOVA. The data underlying this figure are available at https://figshare.com/articles/dataset/Homeostatic_scaling_is_driven_by_a_translation-dependent_degradation_axis_that_recruits_miRISC_remodeling/16768816. mTORC1, mammalian Target Of Rapamycin Complex-1; ns, not significant.

Our observations echo previous findings in Hebbian plasticity, wherein protein synthesis during LTP/LTD was required to counter the changes in the proteomic pool triggered by protein degradation. The blockade of L-LTP accomplished by inhibiting protein synthesis was revoked on the simultaneous application of proteasomal blockers and translational inhibitors [43]. Abrogation of proteasomal activity allowed mGluR-dependent LTD to proceed when protein synthesis was coinhibited [44]. These observations emphasize the existence of a proteostasis network that enables compositional changes to the proteome in contexts of acute or chronic changes in synaptic function [32,45,46]. As LTP and LTD modify the cellular proteome through the simultaneous recruitment of protein synthesis and degradation, it stands to reason that homeostatic scaling mechanisms may also employ a functional synergy of the two to recompense for the changes brought about by unconstrained Hebbian processes.

AMPAR-mediated currents decrease more than NMDAR currents during chronic network hyperactivity [14,18], and, unlike NMDARs, the turnover of AMPARs is translation dependent [47]. The reduced level of sAMPARs following chronic hyperactivity is reset to basal level by the coapplication of protein synthesis and proteasome inhibitors, suggesting that the combined action of translation and degradation affects postsynaptic scaling specifically through

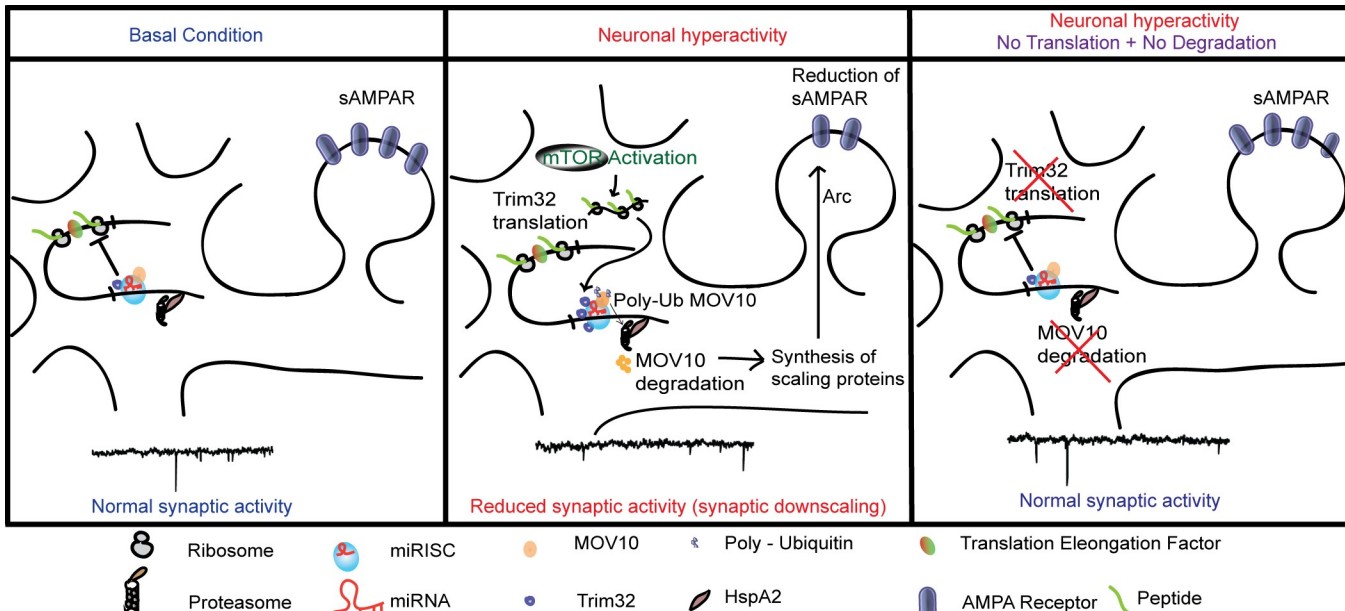

**Fig 14. Schematic representation showing the regulation of homeostatic synaptic activity by the coordinated control of protein synthesis and degradation that modulates miRISC composition.** miRISC, miRNA-induced silencing complex; sAMPAR, surface AMPAR.

sAMPARs. Similar to the observations in synaptic upscaling [13], restricting changes to the sAMPAR abundance by the inhibition of GluA2-endocytosis using GluA2$_{3Y}$ peptide also blocks synaptic downscaling (Fig 2), reinforcing that AMPARs indeed remain the end-point effectors despite changes to the proteome.

## Association of the translation and degradation apparatus is RNA dependent

The colocalization of polyribosomes and proteasomes in neuronal subcompartments suggests that for translation and proteasomal degradation to work in tandem, physical proximity between the two modules cannot be ruled out [30,31]. Polysome analysis showed the cosedimentation of members of the 19S proteasome (Rpt1, Rpt3, and Rpt6 subunits) and the 20S proteasome (α7 subunit) along with translation initiation factors such as eIF4E and p70 S6K, a downstream effector of mTORC1. Abrogation of the sedimentation pattern in the presence of RNase or EDTA is indicative of an RNA-dependent direct interaction between translation and protein degradation (Fig 3). RNase treatment also abolished the direct interaction between proteasome subunits and translation regulators within the polysome, suggesting that actively translating transcripts act as a scaffold to link the translation and proteasome machineries (Fig 4). Such existence of direct interaction between polyribosomes and catalytically active proteasomes allows close temporal coordination between translation and protein degradation.

Does chronic hyperactivity influence the association of proteasome and translation regulators within the polysome? The bicuculline-induced enrichment of the 26S proteasome, phosphorylated S6, p70 S6K and its phosphorylated form, and eEF2 in polysomes is indicative of activity-dependent proximity between the translation and degradation machineries (Fig 5). Interestingly, eEF2 has been previously characterized as a biochemical sensor for synaptic scaling [10]. Trim32, an E3 ligase, was enriched in polysomes, whereas MOV10 and Ago were depleted from polysomes on chronic bicuculline treatment. Trim32 and MOV10 were also present in polysomes in basal conditions (Fig 5).

Previously, MOV10 and Trim32 have been implicated in miRISC-independent functions to modulate RNA modification [48], stability [49], and transcription [50], respectively. Density gradient fractionations of cytoplasmic lysates obtained from nonneuronal systems have revealed that Ago cosedimented with miRNAs in polysome fractions [51,52]. MOV10 was also found associated with polysomes [24]. Interestingly, Trim32 and Ago coimmunoprecipitated with MOV10 from neurons, suggesting that they closely interact with each other and are members of the miRISC (Fig 4F). Apart from this observation, polysome association of Trim32, MOV10, and Ago is bicuculline responsive (Fig 5). We found that chronic bicuculline stimulation triggers a change in the association of Trim32 and MOV10 with Ago (Fig 6). Evidence of their direct physical interaction led us to infer that both MOV10 and Trim32 are part of the miRISC. Hence, association of Ago, Trim32, and MOV10 with polysome can be representative of the association of the miRISC with polysomes, at least in the context of synaptic downscaling.

How does the proteasome remain associated with actively translating mRNAs? We have identified that HspA2 (Hsp70 family), a chaperone protein, remains tethered to proteasomes and polysomes (Figs 4 and 5). Hsp70 family of proteins is known to influence both the synthesis and degradation of proteins by their association with 26S proteasomal subunits [53] and translation initiation factors [54]. HspA2 has been shown to be an interacting partner of the miRISC [55]. Therefore, HspA2 could potentially function as a proteostasis coordinator, which includes members of the proteasome, translation regulators, and chaperone proteins.

## Bicuculline-mediated regulation of Trim32 and MOV10 causes miRISC remodeling during downscaling

We found that during chronic bicuculline treatment, translation of Trim32 precedes the degradation of MOV10. The alternative possibility that MOV10 degradation leads to increased *de novo* translation of Trim32 is not supported, since protein synthesis inhibition by anisomycin leads to MOV10 rescue (Fig 6). Loss of Trim32 prevented bicuculline-induced polyubiquitination and subsequent degradation of MOV10, suggesting that Trim32 is the only E3 ligase marking MOV10 for degradation during synaptic scaling (Fig 6). Immunoprecipitation of Ago from cultured neurons following chronic bicuculline treatment showed MOV10 to be depleted and Trim32 to be enriched, without any change in Dicer expression (Fig 6). Previously, changes to the components of the miRISC have been termed as miRISC remodeling [25], and such compositional changes result in the alteration of miRISC activity [23,56]. Our observations therefore emphasize that such compositional changes within the miRISC, or miRISC remodeling, are effectuated through the reciprocal translational degradation of the Trim32-MOV10 axis. Prolonged bicuculline treatment also did not influence the Ago level, indicating that specific components of the silencing complex are targeted during scaling (Figs 6 and S3).

A recent study has demonstrated that a slow turnover of plasticity proteins (measured at 1, 3, and 7 days in cultured neurons) is essential to create long-term changes to the neuronal proteome during both up- and downscaling [57]. The authors have argued that the slow turnover rate is more energy saving and, therefore, a preferred cellular mechanism. At the same time, this study also identifies a very small fraction of previously reported scaling factors with fast turnover rates specifically influencing up- and downscaling. Our reports support the latter findings, where we observe that both the increase in Trim32 synthesis and the resultant degradation of MOV10 happen within 24 hours during synaptic downscaling, suggesting a fast turnover. As both MOV10 and Trim32 are part of the miRISC, their fast turnover rates seem plausible, considering that participation of the miRISC is mandatory to relieve the translational repression of several transcripts encoding plasticity proteins and needs to happen rapidly in order to boost changes to the proteome. Although in terms of energy expenditure the coordinated regulation of translation and degradation is expensive, this cellular trade-off may be necessary to trigger the remodeling of a very limited number of master regulators of the neuronal proteome, such as miRISC, during synaptic downscaling.

## mTORC1-dependent Trim32 translation-driven MOV10 degradation is sufficient to cause synaptic downscaling

What does postsynaptic signaling cascade trigger Trim32 translation? We find that chronic bicuculline induction triggers the mTORC1-dependent synthesis of Trim32 with the consequent degradation of MOV10 that is a prerequisite for bicuculline-induced downscaling (Fig 7). Although both p70 S6K and 4E-BP2 are downstream effectors of mTORC1, we find that the phosphorylation of p70 S6K exclusively drives Trim32 translation (Fig 8).

After investigating the upstream regulators of the Trim32-MOV10 axis, we focused on identifying the downstream factors that lead to loss of sAMPAR abundance during downscaling. Most studies have determined the influence of single miRNAs in regulating AMPAR distribution during scaling; however, they have been inadequate in providing a holistic view of the miRNA-mediated control of sAMPAR abundance [19–22]. miRNA function has been shown to be directly correlated with miRISC activity [23,56]. Hence, we explored how miRISC remodeling contributes to synaptic downscaling *via* the regulation of sAMPARs. We found that loss of MOV10 function single-handedly accounted for the loss of sGluA1/A2,

accompanied by a commensurate decrease in mEPSC amplitude under basal conditions, effectively recapitulating the postsynaptic events during downscaling (Fig 9). MOV10 knockdown did not reduce mEPSC amplitude further following bicuculline treatment (Fig 10). This observation indicates that the loss of MOV10 sets the threshold point of synaptic strength and chronic hyperactivity cannot override this set point.

We genetically manipulated MOV10 and Trim32 so that their expression becomes opposite to that observed during scaling, i.e., we overexpressed MOV10 and performed Trim32 RNAi. We found that MOV10 overexpression led to a partial occlusion of downscaling (Fig 10). Though myc-tagged MOV10 was amenable for degradation by UPS, yet sustained increased levels of the ectopically expressed MOV10 even after bicuculline treatment was sufficient to cause partial impairment of downscaling. However, chronic bicuculline treatment post-Trim32 RNAi resulted in an increase in mEPSC amplitudes in neurons, which is commensurate with the enhancement of sAMPARs (Figs 11 and 12). Thus, Trim32 knockdown caused a partial impairment of downscaling.

We also observed that Trim32 RNAi under basal conditions causes a modest but statistically significant increase in mEPSC amplitude (Fig 11). We anticipate that this could be due to enhanced MOV10 expression following Trim32 knockdown under basal condition (Fig 6E).

Since the reversal of Trim32 and MOV10 expression levels lead to the partial abolishment of downscaling in neurons upon chronic bicuculline treatment, we infer that the Trim32-MOV10-mediated miRISC remodeling is pivotally positioned to regulate synaptic scaling.

## mTORC1 triggers miRISC remodeling to regulate Arc synthesis during downscaling

Similar to a previous observation [38], our study shows that bicuculline-induced hyperactivity enhances Arc protein, a known regulator of AMPAR removal from synapses (Fig 13). Enhanced Arc translation during chronic bicuculline treatment was blocked by rapamycin, suggesting a regulatory role of mTORC1 in its expression. Furthermore, increase in Arc translation (Fig 13) and concomitant reduction of sAMPARs after loss of MOV10 (Fig 9) demonstrates Arc to be a crucial intermediate between MOV10 degradation and synaptic downscaling.

How is Arc expression regulated during chronic hyperactivity? Our data demonstrate that in the context of downscaling, rapamycin-sensitive Arc expression is driven by posttranscriptional control rather than transcriptional regulation. Chronic hyperactivity-dependent enhanced expression of Arc requires the 3′ UTR of the transcript containing multiple miRNA-binding sites [40], suggesting an involvement of miRISC in scaling. Trim32 knockdown or mTORC1 inhibition prevents the bicuculline-induced increase in Arc reporter expression. We observed that loss of Trim32 alone does not affect the reporter activity, but MOV10 knockdown is sufficient to enhance it (Fig 13). Since MOV10 RNAi mimics the bicuculline-induced reduction of MOV10 during synaptic hyperactivity, and MOV10 regulates miRISC function, we believe that miRISC remodeling is a key determinant of Arc expression. To further enquire whether change of miRISC activity is sufficient to cause increased Arc expression, we performed the RNAi of Dicer, another key component of the miRISC. Similar to MOV10, loss of Dicer has been shown to inhibit miRISC function [56]. Enhanced Arc reporter activity upon Dicer knockdown emphasizes the requirement of miRISC function in regulating Arc (Fig 13). Arc mRNA levels remain unaffected upon chronic hyperactivity in presence or absence of rapamycin as well as knockdown of MOV10 (Fig 13), suggesting that transcription does not play a role in regulating Arc in the context of bicuculline-induced downscaling.

In contrast to chronic hyperactivity-driven loss of MOV10, its polyubiquitination and subsequent localized degradation at active synapses has been shown to occur within minutes upon

glutamate stimulation of hippocampal neurons in culture, or during fear memory formation in amygdala [23,58]. These observations indicate that MOV10 degradation is a common player involved in both Hebbian and homeostatic forms of plasticity. Hebbian plasticity paradigms trigger homeostatic scaling in neurons as a compensatory mechanism [5]; these two opposing forms of plasticity therefore must involve a combination of overlapping and distinct molecular players. Our data demonstrate the requirement of a rapamycin-sensitive, MOV10 degradation-dependent Arc translation in homeostatic scaling that is distinct from the rapamycin-insensitive dendritic translation of Arc occurring during Hebbian plasticity [59]. We speculate that homeostatic and Hebbian plasticity engages distinct signaling pathways that converge at miRISC remodeling.

Though most homeostatic scaling studies including ours used hippocampal neurons in culture to investigate the mechanistic details, the use of this model leaves a lacuna to evaluate how input-specific gene expression control at selective synapses during Hebbian plasticity influences compensatory changes across all synaptic inputs to achieve network homeostasis. Therefore, physiological relevance of homeostatic scaling needs to be studied in association with Hebbian plasticity in order to delineate factors contributing to proteostasis involving cell intrinsic and extrinsic variables within a circuit.

## Methods

### Ethics statement

All animal handling and associated procedures were approved by the IAEC (Institutional Animal Ethics Committee) of NBRC (National Brain Research Centre), India, according to the protocol numbers (NBRC/IAEC/2015/105; NBRC/IAEC/2018/141; NBRC/IAEC/2020/172). The IAEC of NBRC is registered with the CPCSEA (Committee for the Purpose of Control and Supervision of Experiments on Animals) (Registration number: 464/GO/ReBi-S/Re-L/01/CPCSEA) and follows the guidelines of the CPCSEA under the Ministry of Fisheries, Animal Husbandry and Dairying, Government of India.

### Animals

All animals were handled according to the guidelines mentioned by the NBRC-IAEC. Animals were individually housed in cages with food water *ad libitum* and kept on a 12-hour light/dark cycle. Ambient temperature and humidity were maintained at $22 \pm 0.5°C$ and at $40 \pm 5\%$, respectively.

### Primary neuronal culture

Hippocampal neuronal cultures from rat (Sprague Dawley (SD)) were prepared and maintained as described previously [60]. Briefly, hippocampi from embryonic day 18 (E18) pups were dissected, treated with trypsin (0.25%), dissociated by trituration to make single-cell suspension, and plated onto poly-L-lysine (1 mg/mL, Sigma) coated glass coverslip (160 to 250 cells/mm$^2$). About 160 to 170 cells/mm$^2$ were used for electrophysiology and surface labeling experiments. About 200 to 250 cells/mm$^2$ were used for all biochemical experiments. Cortical neuronal cultures from rat were prepared following previous protocol [23]. For polysome experiments, 450 to 475 cortical cells/mm$^2$ were plated onto 90 mm dishes precoated with 1 mg/mL poly-L-Lysine (Sigma). Neurons were maintained in Neurobasal medium (Gibco) containing B27 supplements (Gibco) at 5% $CO_2$/37°C up to 22 to 25 days prior to commencement of experiments. Animal experiments were performed with the approval of the Institutional Animal Ethics (IAEC) committee of National Brain Research Centre.

## Pharmacological inhibitors

Primary hippocampal neurons aged DIV 21 to 24 were treated with bicuculline (10 μM, Tocris), lactacystin (10 μM, AM Systems), and anisomycin (40 μM, Sigma) alone or in combination for 24 hours before further analysis. For mTORC1 inhibition experiments, LY2584702 Tosylate, a competitive inhibitor of p70 S6K was added (final concentration 2 μM) to cultured hippocampal neurons for 24 hours. See also Table 1.

## Lentivirus production and transduction

Lentivirus preparations and transduction into hippocampal neuronal cultures were performed as described previously [23]. Validated shRNA against Trim32 (TATACCTTGCCTGAAG ATC) [33] or Dicer 1 (GCATGGTGGTGTCGATATT) [61] was cloned into MluI and ClaI sites of the pLVTHM vector (Addgene) and verified by sequencing. pLVTHM vectors containing MOV10 shRNA cassettes (sh#1:TTATACAAGGAGTTGTAGGTG) or (sh#2: ACTTAGC TCTAGTTCATAACC) [23] and a nontargeting control (ATCTCGCTTGGGCGAGAGTA AG) were used for lentivirus preparation. Lentivirus particles were produced by cotransfection of 20 μg transfer vector (EGFP cassette under EF1α promoter and shRNA cassette against MOV10 or Trim32 or nontargeting control under H1 promoter in pLVTHM plasmid), 15 μg psPAX2, and 6 μg pMD2.G into HEK293T cells. The cells were grown in low glucose DMEM media (Gibco) with 10% fetal bovine serum (Gibco) and maintained at 5% $CO_2$/37˚C. HEK293T ($2 \times 10^6$ cells) were transfected by calcium phosphate method. Culture supernatant containing lentivirus particles were collected 72 hours posttransfection, and concentrated virus stock was prepared by ultracentrifugation and viral titers determined.

To perform RNAi, hippocampal neurons at DIV 14 to 15 were infected with lentivirus expressing shRNAs against MOV10, Trim32, Dicer, and nontargeting control as mentioned. Viral infections were performed at MOI of 1 to 2 for 6 hours, and following infection lentivirus containing media was replaced with fresh Neurobasal media with B27 supplements. Transduced neurons were incubated up to DIV 23 to 25 with bicuculline (where mentioned) prior to surface labeling and biochemical experiments. Viral infected neurons were tracked by EGFP expression for electrophysiology and imaging experiments.

## Surface labeling of GluA1/A2

Surface expression of AMPAR subunits (GluA1 or GluA2) was analyzed by live labeling of hippocampal neurons with primary antibodies against surface epitopes of GluA1 (Millipore) or GluA2 (Millipore), under different conditions. Neurons (DIV 21 to 24) were immunostained as described previously [62]. Prior to immunostaining, neurons were transduced with lentivirus for protein knockdown or treated with vehicle (DMSO), bicuculline (10 μM), alone or in combination with lactacystin (10 μM) and anisomycin (40 μM) for 24 hours. Live neurons were incubated for 15 minutes at 5% $CO_2$/37˚C with N-terminus specific mouse GluA1 (1:25) or mouse GluA2 (1:10) antibodies diluted in Neurobasal media containing B27 supplements (Gibco). Following incubation, the cells were washed twice with phosphate buffered saline containing $Mg^{2+}$ and $Ca^{2+}$ (PBS-MC; 137 mM NaCl, 2.7 mM KCl, 10 mM $Na_2HPO_4$, 2 mM $KH_2PO_4$, 1 mM $MgCl_2$, and 0.1 mM $CaCl_2$). Cells were then fixed in PBS-MC containing 2% paraformaldehyde and 2% sucrose for 20 minutes at 37˚C, washed 3 times in PBS-MC at room temperature and blocked with PBS-MC containing 2% BSA for 30 minutes at room temperature. Cells were incubated with Alexa-546 conjugated goat-anti-mouse secondary antibody (1:200, Invitrogen) at room temperature for 60 minutes in blocking solution. Cells were permeabilized with PBS-MC containing 0.1% Triton-X-100 at room temperature for 5 minutes. Cells were further incubated with blocking solution for 60 minutes and then with goat PSD95

**Table 1. Reagents and resources.**

| REAGENT or RESOURCE | SOURCE | CATALOGUE NUMBER |
|---|---|---|
| *Antibodies* | | |
| Anti-GluR1-NT (N-terminus) Antibody, clone RH95 | Merck Millipore | Cat#MAB 2263; |
| Anti-GluR2 Antibody, clone 14C12.2 | Merck Millipore | Cat#MABN1189; |
| Anti-PSD95 antibody | Abcam | Cat#ab12093; |
| eIF4E (C46H6) Rabbit mAb | Cell Signaling Technology | Cat#2067; |
| p70 S6 kinase (49D7) Rabbit mAb | Cell Signaling Technology | Cat#2708 |
| Proteasome 19S Rpt1/S7 subunit monoclonal antibody (MSS1-104) | Enzo Life Sciences | Cat#BML-PW8825; |
| Proteasome 19S ATPase subunit Rpt6 monoclonal antibody (p45-110) | Enzo Life Sciences | Cat#BML-PW9265; |
| Proteasome 19S Rpt3/S6b subunit polyclonal antibody | Enzo Life Sciences | Cat#BML-PW8250; |
| Proteasome 20S core subunits polyclonal antibody | Enzo Life Sciences | Cat#BML-PW8155; |
| Purified anti-HA.11 Epitope Tag Antibody | BioLegend | Cat#901501; |
| Anti-Hsp72 Antibody, clone 3G7 | Merck Millipore | Cat#MABE973; |
| Anti-pan Ago Antibody, clone 2A8 | Merck Millipore | Cat#MABE56; |
| eEF2 antibody | Cell Signaling Technology | Cat#2332; |
| Phospho-p70 S6 kinase (Thr389) (1A5) | Cell Signaling Technology | Cat#9206; |
| Anti-TRIM32 antibody | Abcam | Cat#ab96612; |
| MOV10 Antibody | Bethyl Lab | Cat#A301-571A; |
| Monoclonal Anti-β-Tubulin III (neuronal) antibody (Tuj1) | Sigma Aldrich (Merck) | Cat#T8578; |
| GAPDH | Sigma Aldrich (Merck) | Cat#G9545; |
| Anti-HA.11 Epitope Tag Affinity Matrix | BioLegend | Cat#900801; |
| Donkey anti-Goat IgG (H+L) Secondary Antibody, Alexa Fluor 488 | Thermo Fisher Scientific (Invitrogen) | Cat#A11055; |
| Donkey anti-Goat IgG (H+L) Secondary Antibody, Alexa Fluor 633 | Thermo Fisher Scientific (Invitrogen) | Cat#A21082; |
| Goat anti-Mouse IgG (H+L) Secondary Antibody, Alexa Fluor 546 | Thermo Fisher Scientific (Invitrogen) | Cat#A11030; |
| Goat anti-Rat IgG (H+L) Secondary Antibody, HRP | Invitrogen | Cat#31470; |
| Peroxidase AffiniPure Goat Anti-Mouse IgG (H+L) | Jackson ImmunoResearch | Cat#115-035-003; |
| Peroxidase AffiniPure Goat Anti-Rabbit IgG (H+L) | Jackson ImmunoResearch | Cat#111-035-003; |
| AffiniPure Goat Anti-Mouse IgG (H+L) | Jackson ImmunoResearch | Cat#115-005-062; |
| Rabbit IgG Isotype Control | Invitrogen | Cat#10500C; |
| FK1 antibody | Enzo Life Sciences | Cat# BML-PW8805 |
| 20S Proteasome core subunits | Enzo Life Sciences | Cat# BML-PW8155 |
| Anti-pan Ago Antibody, clone 2A8 | Millipore | Cat# MABE56 |
| Anti-Dicer clone N167/7 | Neuromab | Cat# 75–196 |
| *Chemicals, Peptides, and Recombinant Proteins* | | |
| (−)-Bicuculline methochloride | Tocris Bioscience | Cat#0131; |
| Lactacystin | AG Scientific | Cat#SKU L-1147; |
| Tetrodotoxin | Abcam | Cat#ab120054; |
| Glutamate Receptor Endocytosis Inhibitor, GluR23y, YKEGYNVYG | AnaSpec | Cat#AS-62547; |
| Anisomycin from *Streptomyces griseolus* | Sigma-Aldrich | Cat#A-9789; |
| Rapamycin from *Streptomyces hygroscopius* | Sigma-Aldrich | Cat#R-0395; |
| Recombinant Protein G Agarose | Invitrogen | Cat# 15920010 |
| EDTA-free Protease Inhibitor Cocktail | Roche (Sigma Aldrich) | Cat#05892791001 |
| LY2584702 tosylate | Sigma | Cat# SML2892 |

*(Continued)*

**Table 1.** (Continued)

| REAGENT or RESOURCE | SOURCE | CATALOGUE NUMBER |
|---|---|---|
| Cycloheximide | Sigma Aldrich | Cat# C1988; |
| SUPERaseIn RNase Inhibitor (20 U/µL) | Invitrogen (Ambion) | Cat# AM2694 |
| Phospahatase Inhibitor Cocktail 1 | Sigma Aldrich | Cat# P2850 |
| RNase A | Thermo Fisher Scientific (Ambion) | Cat# AM2269 |
| Rnase T1 | Thermo Fisher Scientific (Ambion) | Cat# AM2283 |
| *Critical Commercial Assays* | | |
| BCA protein assay kit | Thermo Fisher Scientific (Pierce) | Cat#23227; |
| Clean-Blot IP Detection Kit (HRP) | Thermo Fisher Scientific (Pierce) | Cat#21232; |
| Immobilon western chemiluminiscent HRP substrate | Millipore | Cat#WBKLS0500 |
| 20S Proteasome Assay Kit | Enzo Life Sciences | Cat# BML-AK740 |
| Dual luciferase Assay Kit | Promega | Cat# E1910 |
| SuperScript III First-Strand Synthesis System | Thermo Fisher Scientific | Cat# 18080051 |
| *Experimental Models: Cell Lines* | | |
| HEK293T | ATCC | ATCC #CRL-3216 |
| *Experimental Models: Organisms/Strains* | | |
| Sprague Dawley Rats | National Brain Research Centre, India | N/A |
| Ribo Tag transgenic mice in C57BL6/J background | The Jackson Laboratory | 029977 |
| T29-1 CamK2α-Cre transgenic mice in C57BL6/J background | The Jackson Laboratory | 005359 |
| *Recombinant DNA* | | |
| pLVTHM (used for preparing lentivirus) | Addgene | Plasmid #12247 |
| psPAX2 | Addgene | Plasmid #12260 |
| pMD2.G | Addgene | Plasmid #12259 |
| *Software and Algorithms* | | |
| ImageJ | NIH | RRID: SCR_003070;https://imagej.net/ |
| GraphPad Prism8 | GraphPad Software | RRID: SCR_002798; http://www.graphpad.com/ |
| MATLAB | The MathWorks | RRID: SCR_001622;http://www.mathworks.com/products/matlab/ |
| pCLAMP10.5 | Molecular Devices | RRID: SCR_011323;http://www.moleculardevices.com/products/software/pclamp.html |

antibody (1:200, Abcam) for 8 hours at 4°C. Cells were incubated with Alexa-633 or Alexa-488 conjugated donkey-anti-goat secondary antibody (1:200, Invitrogen) at room temperature for 90 minutes. Cells were washed 3 times with PBS-MC at room temperature and mounted on Vectashield mounting media with DAPI (Vector Laboratories). See also Table 1.

## Confocal imaging and image analysis

sAMPAR subunits on hippocampal neurons following MOV10 knockdown or bicuculline treatment were imaged using a Leica TCS SP8 point scanning confocal microscope with a Leica Plan Apochromat 63X NA = 1.4 oil immersion objective at 1024 × 1024 pixel resolution. GFP and Alexa 488 were excited by 488 nm Argon laser. Alexa 546 and Alexa 633 were excited by solid state and Helium-Neon lasers, respectively. GFP, Alexa 488, and Alexa 546 signals were detected by hybrid detectors and Alexa 633 was detected by PMT. All images (8 bit) were acquired with identical settings for laser power, detector gain, and pinhole diameter for each experiment and between experiments.

sAMPARs for neurons with Trim32 knockdown in the presence or absence of bicuculline were imaged using a Nikon A1 HD25 point scanning confocal microscope with a Nikon Plan Apochromat 100X NA = 1.4 oil immersion objective at $1024 \times 1024$ pixel resolution. High-magnification images were captured using 2X optical zoom. We obtained 4 to 6 optical sections with 0.5 μM step size. GFP were excited by 488 nm solid-state laser. Alexa 546 and Alexa 633 were excited by solid state lasers. GFP and Alexa 546 were detected by GaAsP detectors. Alexa 633 was detected by PMT. All images (16 bit) were acquired under identical conditions of laser power, detector gain, and pinhole diameter throughout.

High-magnification images, captured from confocal microscopy, were analyzed to observe the intensity of GluA1/A2 expression colocalizing with PSD95 (and GFP for MOV10 RNAi experiments). Images from the different channels were stacked and projected at maximum intensity using ImageJ (NIH). These images were then analyzed using custom written Matlab (MathWorks) programs. First, PSD95 and GFP image signals were thresholded to identify the pixels expressing PSD95 and GFP. Then, the pixels of GluA1/A2, colocalizing with PSD-95 and/or GFP, were filtered, and the average global intensity of these colocalizing GluA1 pixels were collected, plotted, and further analyzed for statistics.

## Polysome fractionation and TCA precipitation of polysome fractions

**Polysome fractionation from rat hippocampus.**   Polysomes from the hippocampi of 8- to 10-week-old SD rats were analyzed following previous protocol [63]. Following decapitation, the brains were removed and placed in ice-cold HEPES HBSS (1× Hank's balanced salt solution, 2.5 mM HEPES-KOH (pH 7.4), 35 mM glucose, and 4 mM NaHCO3) containing 100 μg/mL of cycloheximide. From this point on, all experimental steps were done at 4˚C. Hippocampi were dissected, pooled, and homogenized in homogenization buffer (10 mM HEPES-KOH (pH 7.4), 150 mM KCl, 5 mM $MgCl_2$, and 0.5 mM DTT) containing EDTA-free complete protease inhibitors. A volume of 1.2 mL of homogenization buffer per 4 hippocampi was used. Tissues were homogenized manually with a Dounce homogenizer, and the homogenate was spun at $2,000 \times g$, 10 min at 4˚C to discard nucleus. The supernatant (S1) was collected, and NP-40 was added to a final concentration of 1% v/v. After 5 min of incubation on ice, S1 was spun at 20,000g for 10 minutes, the resultant supernatant (S2) was loaded onto a 20% to 50% w/w linear sucrose density gradient (Sucrose buffer: 10 mM HEPES-KOH (pH 7.4), 150 mM KCl, 5 mM $MgCl_2$). In the indicated conditions, EDTA (30 mM) or a combination of RNase T1 (Ambion, 1,000 U/mL) and RNase A (Ambion, 40 U/mL) was added to S2 and incubated for 10 minutes at room temperature before loading it onto the gradient. The gradients were centrifuged at 40,000g, 2 hours at 4˚C in a Beckman Instruments (Fullerton, CA) SW 41 rotor. Fractions of 0.75 mL volume were collected with continuous monitoring at 254 nm using an ISCO UA-6 UV detector.

**Polysome from HA-Rpl22 mice.**   For polysome from the hippocampi of 8- to 10-week-old HA-Rpl22 transgenic mice, exact protocols as above were followed. Where indicated, tissue lysates were treated with a combination of RNase T1 (Ambion, 1,000 U/mL) and RNase A (40 U/mL) prior to loading onto the linear density sucrose gradient.

**Polysome fractionation from rat cortical cultures.**   Cortical cultures at DIV 21 were incubated with 10 μM bicuculline or vehicle for 24 hours. Polysomes were prepared from cortical cultures following a previous protocol [63] with minor modifications. Seven 90 mm dishes plated with cortical neurons were used in each case. Postincubation, cells were harvested in ice-cold HEPES-HBSS containing 100 μg/mL cycloheximide. Following centrifugation at 3,220g, cells were lysed using homogenization buffer (20 mM HEPES-KOH (pH 7.4), 5 mM $MgCl_2$, 150 mM KCl, 0.5 mM DTT) containing 100 μg/mL cycloheximide, EDTA-free

complete protease inhibitor (Roche, 1 tablet per 5 mL) and 40 U/mL RNase inhibitor (Roche). All steps were carried out at 4˚C. NP-40 was then added to a final concentration of 0.3% of total lysate volume and lysates incubated for 5 minutes on ice, before centrifugation at 12,000$g$ for 30 minutes at 4˚C. Supernatants containing equal amount of protein were loaded onto 20% to 50% w/w sucrose density gradient (described above) and centrifuged at 40,000$g$, 4˚C for 3 hours in SW 41 rotor. The gradients were then fractionated as above. See also Table 1.

**TCA precipitation.**   Sodium dodecyl sulphate was added to a final concentration of 0.015% to each polysome fraction and incubated for 30 minutes in room temperature. Trichloroacetic acid (TCA) was added to polysome fractions at 25% of their volume. All the fractions were incubated for 30 minutes post-TCA addition followed by centrifugation at 13,000$g$ for 30 minutes at room temperature. The pellets were washed with ice-cold acetone (Merck) twice and dried. Acetone residues were allowed to evaporate and the pellets were resuspended in Laemmli buffer.

## Proteasome activity assay

Proteasome activity in monosome and polysome fractions was analyzed by 20S Proteasome Assay Kit (Enzo Life Sciences) as per manufacturer's protocol. Briefly, 20S proteasome chymotrypsin-like activity was tested by incubating 80 µl of each fraction with Suc-LLVY-AMC fluorogenic peptide substrate with or without epoxomicin (500 nM) for 15 minutes at 30˚C. Fluorescence was detected by fluorometer (Tecan). See also Table 1.

## Immunoprecipitation experiments

i. From HA-Rpl22 mice

HA-tagged ribosomes from adult male mice were immunoprecipitated following previous protocol [35] with minor modifications. RiboTag mice were crossed with CamKII-Cre mice and CamKII-Cre:RiboTag offspring expressing HA epitope–tagged Rpl22 were selected by genotyping. Anti-HA-tagged beads (200 µl) were washed twice with citrate–phosphate buffer (pH 5) (24 mM citric acid, 52 mM dibasic sodium phosphate) and allowed to equilibrate twice for 5 minutes each in immunoprecipitation buffer (50 mM Tris (pH 7.5), 100 mM KCl, 12 mM MgCl$_2$, 1% Nonidet P-40). Hippocampi from four adult (8- to 10-week-old) HA-Rpl22 male mice were taken for preparing homogenates, along with the same number of age-matched RiboTag mice who do not express epitope-tagged Rpl22. Hippocampi were rapidly removed and weighed before homogenization in (10% w/vol) polysome buffer (50 mM Tris (pH 7.5), 100 mM KCl, 12 mM MgCl$_2$, 1% Nonidet P-40(NP-40), 1 mM DTT, 100 µg/mL cycloheximide, EDTA free Roche Protease inhibitor cocktail, 200 U/mL RNase Inhibitor) using a Dounce homogenizer. Homogenates were then pelleted at 5,000$g$, 10 minutes at 4˚C followed by collection of supernatant and recentrifugation of the supernatant at 10,000$g$ for 10 minutes at 4˚C to create a postmitochondrial supernatant. The supernatant was precleared with protein-G agarose beads (Invitrogen) for 1 hour, followed by centrifugation at 8,000$g$, 4˚C, for 10 minutes to remove the beads. The precleared supernatant (250 µl) was then incubated with the equilibrated anti-HA-tagged affinity matrix for 6 hours with continuous mixing. The matrix was recovered by centrifugation at 8,000g, 4˚C for 15 minutes followed by 2 washes with high salt buffer HS-150 (Tris 50 mM (pH 7.5), KCl 150 mM, MgCl$_2$ 12 mM, 1% NP-40, DTT 1 mM, 100 µg/mL cycloheximide, protease and RNase inhibitors as above) for 5 minutes and 2 washes with high salt buffer HS-300 (Tris 50 mM (pH 7.5), KCl 300 mM, MgCl$_2$ 12 mM, 1% NP-40, DTT 1 mM, 100 µg/mL cycloheximide, protease and RNase

inhibitors as above) for 5 minutes. All procedures were done at 4˚C. The pellets were boiled in Laemmli buffer, and supernatant was used for analysis.

ii. From polysome fractions

Sucrose gradient fractionations were performed from the hippocampus of mice expressing HA-Rpl22 (see above). High density sucrose fractions from Fraction # 7 to # 15 were pooled from polysomes of each condition (with or without RNase) and diluted using homogenization buffer (see above). Immunoprecipitation from the pooled fractions was performed following a previous protocol [35] using HA antibody-conjugated agarose beads (BioLegend). Incubation with the HA-tagged antibodies was allowed to take place for 18 hours in order to capture all HA-associated protein complexes from the diluted solution. All procedures performed at 4˚C. The immunoprecipitated pellet was washed using a high salt buffer (10 mM HEPES-KOH (pH 7.4), 200 mM KCl, 5 mM $MgCl_2$ along with protease inhibitors) and boiled in Laemmli buffer. The supernatant was used for western blot analysis.

iii. From wild-type Sprague Dawley rats

Immunoprecipitation of 26S proteasome subunits and MOV10 were performed from adult SD rats. Hippocampi of 4 adult (8- to 10-week-old) male SD rats were collected and homogenized in tissue lysis buffer (50 mM Tris (pH 7.5), 150 mM NaCl, 1% NP-40, 2 mM EDTA, Roche protease inhibitor cocktail, 200 U/mL Invitrogen RNase inhibitor, and phosphatase inhibitor cocktail (Sigma)) (10% w/vol) using a Dounce homogenizer. Prior to this, recombinant protein G-agarose beads (Invitrogen) were equilibrated in wash buffer WB-150 (10 mM Tris (pH 8), 150 mM NaCl, and 0.1% NP-40) twice for 5 minutes each and centrifuged at 5,000$g$ for 2 minutes at 4˚C to recover. The homogenates were centrifuged at 2,000$g$, 4˚C for 10 minutes followed by collection of supernatant and recentrifugation at 10,000$g$ at 4˚C for 15 minutes to get a postmitochondrial supernatant. Protein content of the supernatant was measured using the BCA protein estimation method (Pierce). About 2% of the total protein content was kept aside as total input and the remaining was divided into 2 parts having equal protein content (approximately 250 μl each), one to be used for isotype control and the other for experiment purposes. Protein-G agarose (Sigma) beads preblocked with 3% BSA were added (20 μg) to each part and allowed to incubate with continuous mixing at 4˚C for 1 hour. The precleared supernatants were collected by centrifugation at 5,000$g$ for 10 minutes at 4˚C. To the control fraction, 5 μg of IgG isotype control was added (Mouse IgG in case of Rpt6 and Rabbit IgG in case of MOV10 immunoprecipitation). To the experimental fractions, 5 μg of Rpt6 or MOV10 antibody was added, and both fractions were allowed to incubate for 4 hours with continuous mixing. About 40 μg of protein-G agarose beads were added to the fractions and further incubated for 2 hours. The beads were recovered by centrifugation and washed twice with wash buffer IPP-150 (50 mM Tris (pH 7.5), 150 mM NaCl, 12 mM $MgCl_2$, 1% NP-40, and 0.5 mM DTT along with RNase, protease, and phosphatase inhibitors; see reagent list) followed by twice with IPP-300 (same constituents as IPP-150 except NaCl concentration is 300 mM). In case of Rpt6, a further stringent wash with IPP-450 (450 mM NaCl, rest same as IPP-150) was required. All procedures were done at 4˚C. The total input, control, and the immunoprecipitated samples were boiled in Laemmli buffer and stored for further analysis.

iv. From cultured neurons

Ago was immunoprecipitated from primary hippocampal cultures following previous protocol [64]. Cultured hippocampal neurons were incubated on DIV 21 with bicuculline (10 μM) for 24 hours. The following day, cultures were washed with ice-cold 1X PBS with 5 mM $MgCl_2$ and lysed with lysis buffer (20 mM Tris–HCl (pH 7.45), 200 mM NaCl, 2.5 mM

MgCl$_2$, and 1% NP-40) containing EDTA-free complete protease inhibitor and 100 U/mL RNase Inhibitor. Lysates were centrifuged at 16,000$g$, 20 minutes, to remove cellular debris, and supernatants were collected from each condition. About 5 mg of protein-containing lysate from each condition was used for the IP. A volume of 30 μl of protein G-agarose (50% slurry, Sigma) equilibrated with immunoprecipitation buffer (20 mM Tris–HCl (pH 7.5), 200 mM NaCl, and 2.5 mM MgCl$_2$, 40 U/mL RNase Inhibitor and protease inhibitor). Mouse IgG (iso-type control) and equilibrated protein G-agarose beads were used to preclear the lysates for 2 hours. Precleared lysates were incubated with 10 μg of Ago protein (Millipore) overnight with continuous perturbation followed by incubation with the BSA-blocked protein G agarose beads (60 μl of 50% slurry per mL of lysate) for 3 hours at 4˚C. Beads were collected by centrifugation and washed with lysis buffer twice. The beads were resuspended containing 2% SDS and then further incubated with Laemmli buffer at 90˚C. Supernatant from each condition was analyzed by western blot. See also Table 1.

## Polyubiquitination assay and western blot

Rat hippocampal cultures previously transduced with shRNA against Trim32 (Trim32 RNAi) or control shRNA (Control RNAi) were incubated with bicuculline (10 μM) or vehicle for 24 hours on DIV 21. All incubation happened in the presence of lactacystin (10 μM) to capture the spectrum of all polyubiquitinated proteins on bicuculline addition. Following incubation, cells were scraped first with ice-cold phosphate-buffered saline (1X PBS) containing 5 mM MgCl$_2$ and collected by centrifugation at 3,220$g$ at 4˚C for 20 minutes. Cells were then lysed with polysome extraction buffer (PEB) (50 mM Tris–HCl (pH 7.5), 100 mM KCl, 12 mM MgCl$_2$, 1 mM DTT, and 0.1% NP-40) and centrifuged at 16,000$g$, 4˚C for 20 minutes. Protein-G agarose beads were equilibrated with immunoprecipitation (IP) buffer (50 mM Tris–HCl (pH 7.5), 100 mM KCl, 12 mM MgCl$_2$, and 0.05% NP-40). Protein estimation was done, and equal amounts of protein for each condition were taken for further analysis. Lysates were pre-cleared for 2 hours using isotype-specific antibody and protein-G agarose beads. Precleared lysates were then incubated overnight with continuous perturbation at 4˚C with antibody specific for MOV10 (Bethyl Lab). Protein-G agarose beads (50 μl of 50% slurry) were added to each condition and incubated for further 3 hours. Postincubation, the beads were washed twice with high-salt buffer (50 mM Tris–HCl (pH 7.5), 300 mM KCl, 12 mM MgCl$_2$, and 1 mM DTT), followed by 1 wash with PEB. The beads were boiled twice in Laemmli buffer for 10 minutes at 90˚C and the supernatant collected. The supernatants were run on an 8% dena-turing gel and probed with an antibody that recognizes only polyubiquitinylated proteins but not monoubiquitinylated ones (FK1 antibody, 1:1,000, Enzo Life Sciences).

## Western blot

a) From immunoprecipitated samples

Immunoprecipitated samples were analyzed as per previous protocols [23]. Briefly, samples were boiled in Laemmli buffer and equal volumes resolved on 8% to 10% SDS-PAGE. Post-transfer of proteins on nitrocellulose membrane (Millipore), blots were blocked with 5% BSA for 1 hour and probed with primary antibodies overnight. Antibodies used were MOV10 (Bethyl Lab, 1:1,000), Trim32 (Abcam, 1:250), Dicer (NeuroMab, 1:500), Ago (Millipore, 1:1,000), Hspa2 (Millipore, 1:500), Rpt6 (Enzo Life Sciences, 1:500), Rpt1 (Enzo Life Sciences, 1:500), 20S proteasome core (Enzo Life Sciences, 1:500), eEF2 (Cell Signaling Technology, 1:1,000), p70 S6K (Cell Signaling Technology, 1:1,000), eIF4E (Cell Signaling Technology, 1:500), and anti-HA (BioLegend, 1:1000). Following extensive washing with Tris-buffer saline

containing 0.1% Tween-20 (0.1% TBST), secondary antibody supplied with the Clean Blot HRP detection kit (Thermo Fisher Scientific) was used to detect the proteins using standard chemiluminescence detection on X-ray films or the Mini HD9 gel documentation system (UVITEC, Cambridge). Band intensities were quantified by densitometry using ImageJ software.

b) From polysomes

Equal volumes of TCA-precipitated polysome fractions were resolved on 8% to 15% SDS-PAGE and transferred onto nitrocellulose membranes. Following blocking with 5% BSA, blots were probed with Rpt6, Rpt1, Rpt3, 20S proteasome core and α7 subunit of proteasome (Enzo Life Sciences), eIF4E (Cell Signaling Technology), p70 S6K and phospho-p70 S6K (Cell Signaling Technology, 1:500), Rp S6 and phospho-Rp S6 (Cell Signalling Technology, 1:500), Ago (Millipore), Hspa2 (Millipore), Trim32 (Abcam), and MOV10 (Bethyl Lab) overnight at 4˚C. Postincubation, blots were washed with 0.1% TBST and probed with appropriate secondary antibodies. Blots were detected using standard chemiluminescence (Millipore) detection on X-ray films or on the Mini HD9 gel-doc. For polysomes obtained from bicuculline-treated cortical cultures, total band intensity (after background subtraction) of all the bands from fractions 7 to 14 (polysome fractions) were obtained by densitometry on ImageJ and the resultant values normalized with the total area under the curve of all polysome fractions in each polyribosome profile.

c) From cultured neurons

Postincubation with pharmacological inhibitors, cells were washed twice in prewarmed PBS and collected in Laemmli buffer. Equal volumes of lysates were resolved on 8% to 10% SDS-PAGE, transferred onto nitrocellulose membrane, blocked with 5% BSA, and probed with antibodies against MOV10, Trim32, Ago, Dicer, Arg3.1 (CST, 1:250), p70 S6K, and phospho-p70 S6K. Blots were detected using standard ECL chemiluminescence detection (Millipore) and band intensity determined by ImageJ. Blots were normalized to Tuj1. MOV10, Dicer, and Trim32 RNAi samples were also detected similarly. See also Table 1.

## Electrophysiology

Whole-cell patch clamp experiments were performed using primary hippocampal neurons (DIV 18 to 25). Neurons were incubated with bicuculline (10 µM), anisomycin (40 µM), lactacystin (10 µM), rapamycin (100 nM), and GluA23y (10 µM) for 24 hours. Neurons were patched with glass microelectrodes with an open-tip resistance of 3 to 8 MΩ. Cells with series resistance >30 MΩ were excluded from the analysis. To measure the excitatory currents, the following composition of internal solution was used: 100 mM Cesium gluconate, 0.2 mM EGTA, 5 mM MgCl2, 2 mM ATP, 0.3 mM GTP, 40 mM HEPES (pH 7.2) (285 to 290 mOsm). mEPSCs were recorded by holding the cells at −70mV in a recording solution consisting of the following: 119 mM NaCl, 5 mM KCl, 2 mM CaCl2, 2 mM MgCl2, 30 mM glucose, 10 mM HEPES (pH7.4) (310 to 320 mOsm) in the presence of 1 µM tetrodotoxin and 10 µM bicuculline.

Average of mEPSC events for 300 s from each neuron was analyzed, and only the events with <−4 pA of peak amplitudes, >0.3 pA/ms of rise rates, and 1 to 12 ms of decay time constants were selected for the analysis.

All recorded signals were amplified by Multiclamp 700B (Molecular devices), filtered at 10 Khz, and digitized at 10 to 50 KHz. Analog to digital conversion was performed using Digidata 1440A (Molecular Devices). All data were acquired and analyzed using pClamp10.5 software

(Molecular Devices) and custom Matlab filtering algorithms. Cells with holding currents greater than −100 pA were excluded from the analysis, as well as any cell which was unstable during the recording.

## Luciferase assay

Hippocampal neurons transduced with MOV10, Trim32, or Dicer shRNAs at DIV 9 to 10 were transfected at DIV 18 and bicuculline added on DIV 20. Luciferase assays were performed on DIV 21. Cells were cotransfected with Gaussia luciferase (Gluc) containing the complete Arc 3′ UTR from mouse brain cDNA and Firefly luciferase (Fluc) cloned in pMIR-Report (Ambion). Luciferase activities of the two reporters were measured using the Dual Luciferase Reporter Assay System (Promega) according to the manufacturer's instructions. Gluc activity was normalized by Fluc. See also Table 1.

## qRT-PCR analysis of Arc mRNA

Total RNA was isolated from cultured neurons either subjected to MOV10 RNAi or treated with bicuculline and/or rapamycin for 24 hours using Trizol (Invitrogen). cDNA was synthesized using SuperScript III First Strand Synthesis System (Life Technologies, Invitrogen).

Following primers were used for qRT-PCR: Forward: 5′-GGGTGGCTCTGAAGAATATT-3′,
Reverse: 5′-TGTACTGCAGAAACTCCTTC-3′.qRT-PCR results analyzed by δδCt method.
See also Table 1.

## Statistical analysis

Statistical analyses were performed for all experiments. Whole-cell patch clamp amplitudes and frequencies were analyzed using one-way ANOVA with post hoc Fisher's LSD test to test pairwise differences across the groups. Imaging and western blot data were analyzed for statistical significance using one-way ANOVA with post hoc Fisher's LSD test. Western blot data related to RNAi experiment were analyzed using unpaired $t$ test with Welch's correction. Data are reported as absolute differences in mean ± SEM for electrophysiology data or percent differences in mean ± SEM for imaging and western blot data between groups.

## Supporting information

**S1 Fig. Synaptic downscaling by coordinated control of protein synthesis and degradation involves AMPARs. (A, B)** Hippocampal neurons were stained for sGluA1 (A) or sGluA2 (B) and PSD95 as described in Fig 2A and 2B. Photomicrograph showing images for sGluA1 or sGluA2 (red) and PSD95 (green) and sGluA1/PSD95 or sGluA2/PSD95 (merged). High-magnification images of dendrites shown in Fig 2 marked in red square. Scale bar as indicated. Quantitation shown in Fig 2C and 2D. See Fig 2 for data. **(C-E)** mEPSCs traces from hippocampal neurons (DIV 18–24) treated with vehicle or GluA2$_{3y}$ for 24 hours (C) as described in Fig 2E. Scale as indicated. Mean mEPSC amplitudes (D) and frequencies (E) in neurons treated as indicated. $n$ = 12. Data shown as mean ± SEM. One-way ANOVA and Fisher's LSD. See Fig 2 for data. The data underlying this figure are available at https://figshare.com/articles/dataset/Homeostatic_scaling_is_driven_by_a_translation-dependent_degradation_axis_that_recruits_miRISC_remodeling/16768816. AMPAR, AMPA receptor; DIV, days in vitro; mEPSC, miniature excitatory postsynaptic current; ns, not significant.
(TIF)

**S2 Fig. OD$_{254}$ profile of polysome fractionation. (A-G)** A$_{254}$ profile obtained from spectrophotometer attached to gradient fractionator shown in Figs 3 and 4. Traces were drawn from

original $A_{254}$ profile obtained from hippocampal cytoplasmic extract treated with $MgCl_2$ (A), RNase (B), EDTA (C), and $MgCl_2$ (D) shown in Fig 3, $MgCl_2$-treated extract from mouse expressing HA-Rpl22 in excitatory neurons from hippocampus (E) shown in Fig 4C and RNase (F) or without RNase (G)–treated extract from mouse expressing HA-Rpl22 in excitatory neurons from hippocampus shown in Fig 4J. **(H, I)** $A_{254}$ profile of sucrose density fractions obtained from vehicle (H) or bicuculline (I)–treated cortical neurons. Traces were drawn from these original $A_{254}$ profiles as shown in Fig 5A and 5B.
(TIF)

**S3 Fig. Expression profile of miRISC members under basal condition and bicuculline-induced hyperactivity.** Hippocampal neurons (DIV 21) were treated with bicuculline for 24 hours. **(A)** Photomicrograph showing the expression of Ago and Tuj1 as detected by western blot analysis. **(B)** Quantitation of Ago expression. $n = 3$. ns, not significant. Unpaired 2-tailed $t$ test with Welch's correction. **(C-E)** Hippocampal neurons (DIV 21) treated with lactacystin, anisomycin, and both for 24 hours. Photomicrograph showing the expression of Trim32 and MOV10 as detected by western blot analysis (C). Quantitation of Trim32 (D) and MOV10 (E). Data shown as mean ± SEM, $n = 3$, $*p < 0.001$ and $**p < 0.0003$. One-way ANOVA and Fisher's LSD. See Fig 6 for data. The data underlying this figure are available at https://figshare.com/articles/dataset/Homeostatic_scaling_is_driven_by_a_translation-dependent_degradation_axis_that_recruits_miRISC_remodeling/16768816. Ago, Argonaute; DIV, days in vitro; miRISC, miRNA-induced silencing complex; ns, not significant.
(TIF)

**S4 Fig. sAMPARs expression following MOV10 knockdown.** Hippocampal neurons (DIV 14–15) transduced with lentivirus expressing two shRNAs against MOV10 (shRNA#1 or shRNA#2) along with GFP. Transduced neurons (DIV 21–24) were immunostained for surface GluA1 (sGluA1) and coimmunostained for PSD95. **(A)** Photomicrograph showing confocal images of GFP (green), sGluA1 (red), PSD95 (blue), and GFP/sGluA1/PSD95 (merged). High-magnification images of dendrites shown in Fig 9 marked in red square. **(B)** Relative intensity of sGluA1 particles at the synapse (overlap with PSD95 particles onto GFP expressing dendrites). Normalized intensity of sGluA1 relative to control was plotted. Data shown as mean ± SEM. $*p < 0.01$. One-way ANOVA and Fisher's LSD. **(C)** Hippocampal neurons (DIV 14–15) transduced with lentivirus expressing shRNA against MOV10 (shRNA#1) along with GFP. Transduced neurons (DIV 21–24) were immunostained for surface GluA2 (sGluA2) and PSD95. Photomicrograph showing confocal images of GFP (green), sGluA2 (red), PSD95 (blue), and GFP/sGluA2/PSD95 (merged). High-magnification images of dendrites shown in Fig 9 marked in red square. Scale as indicated. Relative intensity of sGluA2 particles at the synapse (overlap with PSD95 particles onto GFP expressing dendrites). Normalized intensity of sGluA2 relative to control was plotted. Data shown as mean ± SEM. $*p < 0.01$. One-way ANOVA and Fisher's LSD. See Fig 9 for data. The data underlying this figure are available at https://figshare.com/articles/dataset/Homeostatic_scaling_is_driven_by_a_translation-dependent_degradation_axis_that_recruits_miRISC_remodeling/16768816. AMPAR, AMPA receptor; DIV, days in vitro; sAMPAR, surface AMPAR.
(TIF)

**S5 Fig. sAMPARs expression in bicuculline-induced neurons following Trim32 knockdown.** Hippocampal neurons (DIV 14–15) transduced with lentivirus expressing shRNA against Trim32 along with GFP. **(A-B)** Transduced neurons (DIV 21–24) were stimulated with bicuculline for 24 hours and immunostained for sGluA1 (A) or sGluA2 (B) and coimmunostained for PSD95. Photomicrograph showing confocal images of GFP (green), sGluA1/

sGluA2 (red), PSD95 (blue), and GFP/sGluA1 or sGluA2/PSD95 (merged). High-magnification images of dendrites shown in Fig 12 A-F marked in red square. Relative intensity of surface GluA1 (A) or surface GluA2 (B) particles at the synapse (overlap with PSD95 particles onto GFP expressing dendrites). Normalized intensity of surface GluA1/GluA2 relative to control was plotted. Data shown as mean ± SEM. $^*p < 0.02$, $^{**}p < 0.03$, for sGluA1. $^*p < 0.008$, $^{**}p < 0.002$, for sGluA2. One-way ANOVA and Fisher's LSD. See Fig 12 for data. The data underlying this figure are available at https://figshare.com/articles/dataset/Homeostatic_scaling_is_driven_by_a_translation-dependent_degradation_axis_that_recruits_miRISC_remodeling/16768816. AMPAR, AMPA receptor; DIV, days in vitro; ns, not significant; sAMPAR, surface AMPAR.
(TIF)

**S6 Fig. Overexpression of MOV10 and knockdown of Dicer. (A)** Myc-tagged MOV10 was transfected in hippocampal neurons (DIV 15) as detected by western blot analysis (DIV 21) using antibody against Myc. See also Fig 10. **(B)** Hippocampal neurons (DIV 14) were transduced with lentivirus expressing shRNA against Dicer or control shRNA. Photomicrograph showing effective knockdown of Dicer (DIV 24) as detected by western blot analysis using antibody against Dicer. See data for Fig 13. The data underlying this figure are available at https://figshare.com/articles/dataset/Homeostatic_scaling_is_driven_by_a_translation-dependent_degradation_axis_that_recruits_miRISC_remodeling/16768816. DIV, days in vitro; IB, immunoblot.
(TIF)

**S1 Data. Excel spreadsheet containing, in separate sheets, the underlying numerical data and statistical analysis for figure panels 1B, 1C, 1E, 1F, 2C, 2D, 2F, 2G, 3H, 5E, 5F, 6B, 6C, 6E, 7B, 7C, 7E, 7F, 8B, 8C, 8E, 8F, 9B, 9D, 9F, 9G, 10B, 10C, 10E, 10F, 11B, 11C, 12E, 12F, 13B, 13D, 13E, 13F, 13G, 13H, and 13I.**
(XLSX)

**S2 Data. Excel spreadsheet containing, in separate sheets, the underlying numerical data and statistical analysis for supporting information figure panels S1D, S1E, S3B, S3D, S3E, and S4B.**
(XLSX)

## Acknowledgments

We thank Dipanjan Roy for helpful discussions, Addgene for lentiviral vectors; Ken Kosik for MOV10 shRNA and myc-MOV10 constructs; Clive Bramham for Arc 3′ UTR fused luciferase reporter construct; microscope facility of Regional Centre for Biotechnology, India; Rohini Roy for Trim32 RNAi construct; Utsav Mukherjee, Premkumar Palanisamy, Gourav Sharma, Anupam Das, and Surajit Chakraborty for technical assistance and animal maintenance; and Karthick Ravichandran and Abir Mondal for helping with polysome fractionation.

## Author Contributions

**Conceptualization:** Sarbani Samaddar, Sourav Banerjee.

**Data curation:** Balakumar Srinivasan, Sarbani Samaddar, Sourav Banerjee.

**Formal analysis:** Balakumar Srinivasan, Sarbani Samaddar, James P. Clement, Sourav Banerjee.

**Funding acquisition:** Sourav Banerjee.

**Investigation:** Balakumar Srinivasan, Sarbani Samaddar, Sourav Banerjee.

**Methodology:** Balakumar Srinivasan, Sarbani Samaddar, James P. Clement, Sourav Banerjee.

**Project administration:** Sourav Banerjee.

**Resources:** Sivaram V. S. Mylavarapu.

**Supervision:** Sourav Banerjee.

**Validation:** Balakumar Srinivasan, Sarbani Samaddar.

**Visualization:** Balakumar Srinivasan, Sarbani Samaddar, Sourav Banerjee.

**Writing – original draft:** Sarbani Samaddar, Sourav Banerjee.

**Writing – review & editing:** Balakumar Srinivasan, Sarbani Samaddar, James P. Clement, Sourav Banerjee.

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
