## [Editor Report · Decision Letter 0]

26 Jul 2021

Dear Dr Banerjee, 

Thank you for submitting your revised manuscript entitled "Homeostatic scaling is driven by a translation-dependent degradation axis that recruits miRISC remodeling" for consideration as a Research Article by PLOS Biology.

Your manuscript has now been evaluated by the PLOS Biology editorial staff and I am writing to let you know that we would like to send your submission out for external peer review.

Please re-submit your manuscript within two working days, i.e. by Jul 28 2021 11:59PM.

Kind regards,

Lucas Smith

Associate Editor

PLOS Biology

lsmith@plos.org

---

## [Decision Letter · Decision Letter 1]

30 Aug 2021

Dear Dr Banerjee,

Thank you for submitting your revised Research Article entitled "Homeostatic scaling is driven by a translation-dependent degradation axis that recruits miRISC remodeling" for publication in PLOS Biology. I have now obtained advice from two of the original reviewers and have discussed their comments with the Academic Editor. 

The reviews are appended below. As you will see, both reviewers are now satisfied by your revision and think the manuscript has been much improved. Based on the reviews, we will probably accept this manuscript for publication, provided you satisfactorily the following data and other policy-related requests:

1) ETHICS REQUESTS: Please include the approval number for your animal care and use protocol, approved by the IAEC committee of National Brain Research Centre. Please also include the specific national or international regulations/guidelines to which your animal care and use protocol adhered. Please note that institutional or accreditation organization guidelines (such as AAALAC) do not meet this requirement.

2) DATA REQUEST: Thank you for providing the underlying data for your study on figshare.com. Please add a reference to this data to each figure legend in your manuscript (including supplemental). For example, to each figure legend you can add the sentence "the data underlying this figure is available at https://figshare.com/articles/dataset/Homeostatic_scaling_is_driven_by_a_translation-dependent_degradation_axis_that_recruits_miRISC_remodeling/15066378"

3) WESTERN BLOT REQUEST: Thank you for providing larger gel images on figshare.com. Looking through these, it appears that some of the western blot images are still cropped (ex Fig 3B). In order to be compliant with our blot and gel reporting requirements, we require the original, uncropped and minimally adjusted images for all blot and gel results reported in an article's figures or Supporting Information files. Please update your data repository to contain the full, uncropped and minimally adjusted images for all western blot data presented in the paper. For more information, please carefully read our guidelines for how to prepare and upload this data: https://journals.plos.org/plosbiology/s/figures#loc-blot-and-gel-reporting-requirements 

4) BLURB REQUEST: Please also provide a blurb which (if accepted) will be included in our weekly and monthly Electronic Table of Contents, sent out to readers of PLOS Biology, and may be used to promote your article in social media. The blurb should be about 30-40 words long and is subject to editorial changes. It should, without exaggeration, entice people to read your manuscript. It should not be redundant with the title and should not contain acronyms or abbreviations. For examples, view our author guidelines: https://journals.plos.org/plosbiology/s/revising-your-manuscript#loc-blurb

5) Please move all methodological information from the supplement into the main text, under the heading “Materials and Methods”

We expect to receive your revised manuscript within two weeks. 

*Published Peer Review History*

*Early Version*

Sincerely,

Lucas Smith, Ph.D.,

Associate Editor,

lsmith@plos.org,

PLOS Biology

Reviewer remarks:

Reviewer #1, Ana Luisa Carvalho: The authors have appropriately addressed the issues that were raised, and the study is now suitable for publication.

Reviewer #2, Yi-Shuian Huang: The authors have appropriately addressed my questions and concerns by performing several key experiments to convincingly demonstrate that the regulatory axis of TRIM32-MOV10-Arc (translation-degradation-miRISC) drives synaptic downscaling. The reviewer thanks the authors' effort to improve this manuscript by solid data and careful interpretation of their experimental results.

---

## [Editor Report · Decision Letter 2]

30 Sep 2021

Dear Dr Banerjee,

On behalf of my colleagues and the Academic Editor, Yi-Ping Hsueh, I am pleased to say that we can in principle offer to publish your Research Article "Homeostatic scaling is driven by a translation-dependent degradation axis that recruits miRISC remodeling" in PLOS Biology, provided you address any remaining formatting and reporting issues. These will be detailed in an email that will follow this letter and that you will usually receive within 2-3 business days, during which time no action is required from you. Please note that we will not be able to formally accept your manuscript and schedule it for publication until you have made the required changes.

PRESS

Sincerely, 

Lucas Smith, Ph.D. 

Senior Editor 

PLOS Biology

lsmith@plos.org